# Mining a human transcriptome database for chemical modulators of NRF2

**John P. Rooney[1,2¤], Brian Chorley[1], Steven Hiemstra[3], Steven Wink[3], Xuting Wang[4], Douglas A. Bell[4], Bob van de Water[3], J. Christopher Corton[1] \***

**1** Center for Computational Toxicology and Exposure, US-EPA, Research Triangle Park, NC, United States of America, **2** ORISE, US-EPA, Research Triangle Park, NC, United States of America, **3** Division of Toxicology, Leiden Academic Center for Drug Research, Leiden University, Leiden, The Netherlands, **4** Immunity, Inflammation, and Disease Laboratory, National Institute of Environmental Health Sciences, Research Triangle Park, NC, United States of America

¤ Current address: ILS, Durham, NC, United States of America
* corton.chris@epa.gov

**Data Availability Statement:** All microarray or RNA-Seq data is available in Gene Expression Omnibus and ArrayExpress. The accession numbers are found in S1 File.

## Abstract

Nuclear factor erythroid-2 related factor 2 (NRF2) encoded by the *NFE2L2* gene is a transcription factor critical for protecting cells from chemically-induced oxidative stress. We developed computational procedures to identify chemical modulators of NRF2 in a large database of human microarray data. A gene expression biomarker was built from statistically-filtered gene lists derived from microarray experiments in primary human hepatocytes and cancer cell lines exposed to NRF2-activating chemicals (oltipraz, sulforaphane, CDDO-Im) or in which the NRF2 suppressor Keap1 was knocked down by siRNA. Directionally consistent biomarker genes were further filtered for those dependent on NRF2 using a microarray dataset from cells after *NFE2L2* siRNA knockdown. The resulting 143-gene biomarker was evaluated as a predictive tool using the correlation-based Running Fisher algorithm. Using 59 gene expression comparisons from chemically-treated cells with known NRF2 activating potential, the biomarker gave a balanced accuracy of 93%. The biomarker was comprised of many well-known NRF2 target genes (*AKR1B10*, *AKR1C1*, *NQO1*, *TXNRD1*, *SRXN1*, *GCLC*, *GCLM*), 69% of which were found to be bound directly by NRF2 using ChIP-Seq. NRF2 activity was assessed across ~9840 microarray comparisons from ~1460 studies examining the effects of ~2260 chemicals in human cell lines. A total of 260 and 43 chemicals were found to activate or suppress NRF2, respectively, most of which have not been previously reported to modulate NRF2 activity. Using a NRF2-responsive reporter gene in HepG2 cells, we confirmed the activity of a set of chemicals predicted using the biomarker. The biomarker will be useful for future gene expression screening studies of environmentally-relevant chemicals.

## Introduction

The vast number of chemicals used in industry have limited toxicity testing. The time and financial costs to screen these chemicals via traditional methods is prohibitive. These constraints have driven the development of high-throughput *in vitro* screening assays to aid in

**Funding:** This research was supported in part by a postdoctoral appointment to JPR to the Research Participation Program for the U.S. Environmental Protection Agency, Office of Research and Development, administered by the Oak Ridge Institute for Science and Education through an interagency agreement between the U.S. Department of Energy and EPA. There was no additional external funding received for this study.

**Competing interests:** The authors have declared that no competing interests exist.

assessing chemical toxicity and to prioritize chemicals for traditional testing. Great strides have been made by United States governmental chemical screening programs including the EPA's ToxCast program and the cross-agency Tox21 program in this regard, screening over 1800 chemicals in as many as 700 assays [1]. Landmark studies have developed computational models that for example predict modulation of estrogen and androgen receptors, important targets for endocrine disrupting chemicals with remarkable accuracy based on results from *in vitro* assays [2–4]. Yet, despite the success of these programs in prioritization, there are still obstacles to overcome. Most HTS assays are targeted to specific events in biological pathways, and while this aids in specificity, it also leaves a large biological space that is not covered by the current assay battery [5, 6]. To more completely assess the effects of chemicals on a wider range of molecular targets of regulatory interest, broader approaches are needed.

Gene expression profiling represents a robust complementary approach to HTS screening and has the potential to cover some of the biological space missed by HTS assays. The Library of Integrated Network-Based Cellular Signatures (LINCS) and the Connectivity Map (CMAP) projects have both made significant contributions to the field of large-scale gene expression profiling, using network-based approaches to link chemical exposure to gene expression and disease. The CMAP project screened ~1300 small molecules in 3 cell lines with whole transcriptome expression data [7, 8]. One of the major hurdles to these early gene expression profiling efforts was the inherently low throughput of microarray technologies. However, the field has seen great advancements in throughput in recent years. The aforementioned LINCS database uses the L1000 gene expression technology that measures the expression of 1000 genes, and through computational inference can predict transcriptional changes in nearly 80% of the genome. The LINCS project has generated over 1 million profiles using the L1000 technology from a large battery of human cell lines perturbed with chemicals and gene probes [9]. Furthermore, new RNA sequencing (RNA-seq) technologies, such as the TempO-Seq platform show great promise in their ability to measure expression changes in both smaller, targeted gene sets, and the whole transcriptome in a high-throughput manner [10]. Lastly, there is a wealth of microarray- and RNA-Seq-derived gene expression data currently available in multiple public repositories that can be used to develop procedures for predicting the molecular targets of chemicals.

A major challenge in the field of gene expression analysis is identifying signals that are indicative of modulation of specific molecular targets. One mechanism by which chemicals can exert toxic effects on a cell is by activating or repressing transcription factors, thus changing gene expression and altering normal cellular signaling events. We have previously developed gene expression biomarkers that can accurately identify chemicals that activate and/or suppress a number of transcription factors in the mouse or rat liver important in cancer and steatosis [11–21]. Our group has also characterized biomarkers that predict modulation of estrogen receptor (ERα) and androgen receptor or predict genotoxicity in human cells [22–24]. These biomarkers consist of short lists of genes (up to ~150) whose expression consistently change as a result of exposure to structurally-diverse chemicals or other perturbants, indicating either activation or suppression of a specific transcription factor. The modulation of the transcription factors that our group has focused on are often molecular initiating events (MIEs) or key events (KEs) in Adverse Outcome Pathways (AOPs). AOPs are defined as a series of mechanistically linked KEs starting with a MIE in which a chemical interacts with a target, culminating in an adverse outcome in a tissue [25]. Thus, gene expression biomarkers can be utilized to interpret microarray profiles with the goal of populating MIE/KE activity in AOP networks [26].

Nuclear factor erythroid-2 related factor 2 (NRF2) encoded by the *NFE2L2* gene is a key transcription factor important in cellular responses to oxidative stress and xenobiotics. Under normal conditions, NRF2 is bound in the cytoplasm by Kelch-like ECH-associated protein 1 (Keap1), which results in ubiquitination and targeting of NRF2 for proteasomal degradation. When activated, NRF2 dissociates from Keap1, translocates to the nucleus and binds to genomic antioxidant response elements (AREs) as a heterodimer with Maf proteins, MafF, MafG, or MafK [27]. NRF2 binding to AREs promotes the transcription of a diverse battery of genes involved in the antioxidant response and detoxification, including many genes related to the cytoprotective processes of phase 2 metabolism [28]. NRF2 is intricately linked with carcinogenesis, as NRF2-nullizygous mice are more susceptible to many chemical carcinogens, yet paradoxically NRF2 and its target genes are also upregulated in many cancers [29, 30]. The activation status of NRF2 is also linked to hepatocyte steatosis [31], a condition in which there is an accumulation of triglycerides. The ability to readily identify chemicals that modulate NRF2 in microarray studies could help to build predictive models for cancer or steatosis.

In the present study, we developed computational methods for using a gene expression biomarker to predict NRF2 activation or suppression in human cells. We used this biomarker, coupled with an annotated database of gene expression profiling experiments, to perform an *in silico* screen for chemical perturbations that lead to NRF2 modulation. We validate our findings using an ARE-linked reporter system in HepG2 cells.

## Methods

### Construction and characterization of the NRF2 biomarker

The overall strategy used to computationally construct and characterize the NRF2 gene expression biomarker is depicted in Fig 1. The biomarker consists of a list of differentially regulated genes whose expression is consistently altered after exposure to chemicals that activate NRF2. Gene expression data were sourced from the commercially available database, called BaseSpace Correlation Engine (BSCE) (https://www.illumina.com/products/by-type/informatics-products/basespace-correlation-engine.html; formally NextBio). Differential gene expression analysis protocols employed in BSCE are described in detail in Kupershmidt et al. [32]. Briefly, raw (or preprocessed data if raw is not available) gene expression data is collected from GEO, ArrayExpress and other public repositories. Expression data are log transformed and normalized as appropriate (RMA, per-chip median or Lowess), and differential expression is calculated using a Welch's or standard t-test, with a p value cut-off of 0.05 without multiple test correction, and a fold change cut-off of +/- 1.2 fold. Genes with expression values in the lower 20th percentile in both groups are removed. Lists of statistically filtered differentially expressed genes resulting from these comparisons are referred to as biosets.

Right, Biomarker Testing and Screening for Modulators. The biomarkers were imported into the BSCE environment and compared to all other human biosets via the Running Fisher algorithm for rank-ordered pairwise comparisons [32]. The p-values and correlation directions were exported and used to populate a master database of experimental details for all biosets. The biomarkers were tested for accuracy by comparing predictions with those from the ToxCast and Tox21 HTS NRF2 activation assays. The biomarker with the best predictive accuracy was used to identify chemicals in the database that activated or suppressed NRF2. Post-hoc analysis of the biomarker gene list included canonical pathway enrichment and ChIP-Seq analysis.

The NRF2 biomarker was constructed from lists of differentially expressed genes derived from chemical and genetic perturbations known to affect the activity of NRF2. The list of biosets used to construct the biomarker are listed in Table 1. Because our goal was to use the

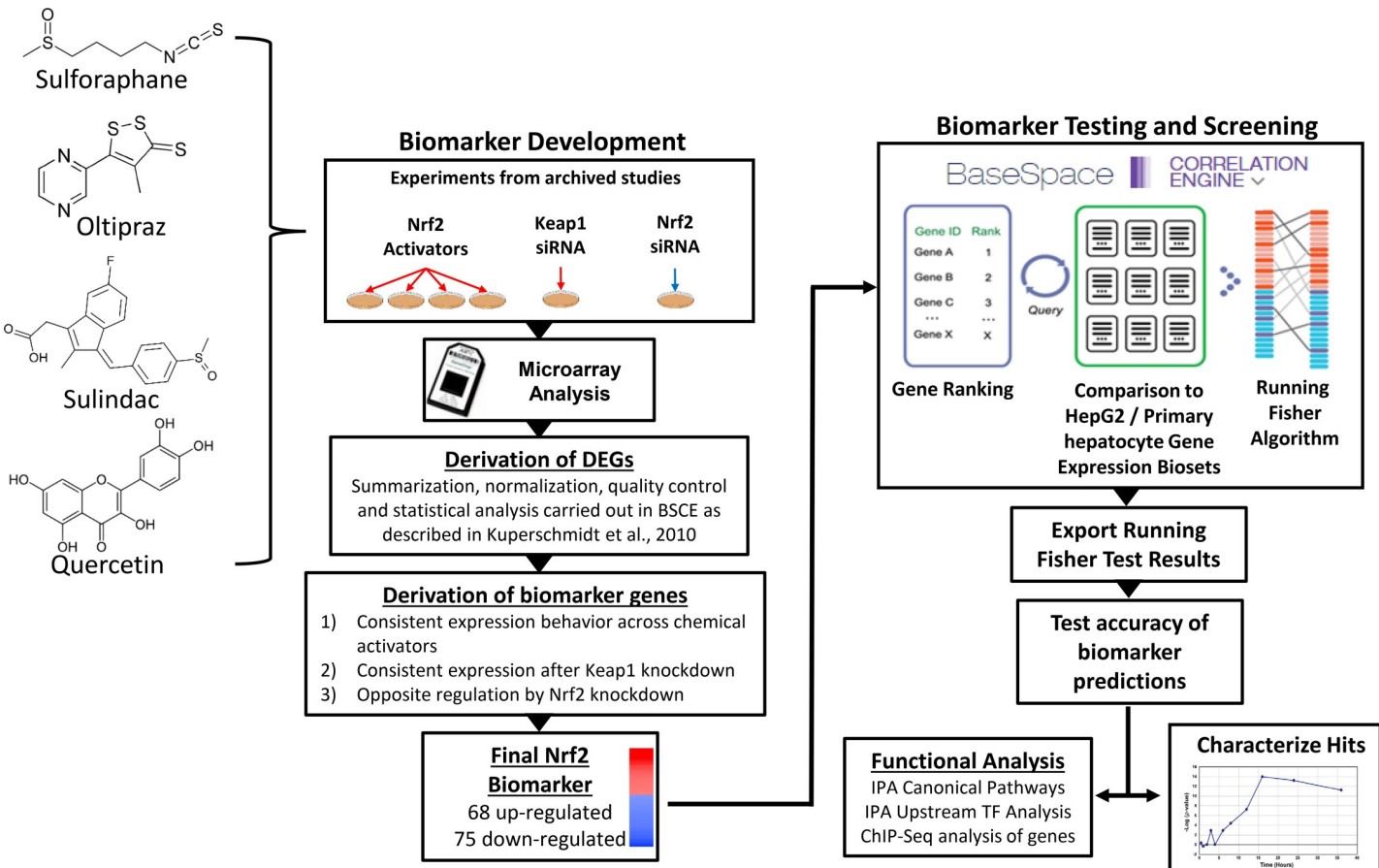

**Fig 1. Biomarker construction and screening strategy.** Left, NRF2 biomarker development. Differentially expressed genes (DEGs) from exposures to known NRF2 activators (sulforafan, oltipraz, sulindac, quercetin) and *KEAP1* and *NFE2L2* siRNA knockdown were assessed in the BSCE environment. Biomarker genes were identified as those consistently up- or down-regulated by the chemical exposures or KEAP1 knockdown and opposing regulation in *NFE2L2* knockdown (further details in Results section). Ten biomarkers were initially created from biosets that varied based on chemical, time and dose of exposure, and tissue context.

biomarker to identify conditions that activate NRF2 in many cell types, we used biosets derived from a variety of different cell lines to find a common set of genes. The biosets were from primary human hepatocytes and human cancer cell lines derived from liver (HepG2), breast (MCF7, MCF10A), and lung (A549). These gene sets were compared to identify those with consistent expression in a series of steps. First, genes were identified that exhibited consistent expression upon exposure to known NRF2-activating chemicals (e.g., sulforaphane, oltipraz, sulindac, quercetin) across the 6 chemical-treated biosets as well as a bioset derived from genetic knockdown of the NRF2 repressor Keap1. To pass this filter genes had to be either up-regulated or down-regulated in all of the biosets in which the gene exhibited differential expression and in the majority of the biosets (4 or more out of 7 biosets). Second, the gene lists were further filtered for those genes that also were regulated in an opposite manner by NRF2 knockdown to filter for direct dependence on NRF2. Any gene that did not meet this criteria was removed. Finally, the genes were filtered for those with robust expression changes. The average fold-change across the 7 biosets in which NRF2 was activated had to be $\geq |+/-$ 1.5-fold| (not Log2(fold-change). The final biomarker consisted of 143 genes. The list of biomarker genes and associated fold-change values is found in S1 File.

**Table 1. Biosets used to build the NRF2 biomarker.**

| Bioset Name | Factor Examined | Cell type | Time of treatment (hr) | Concentration (uM) | Number of Differentially Expressed Genes | Study | NRF2 biomarker [-Log(p-value)] |
|---|---|---|---|---|---|---|---|
| Primary human hepatocytes + 50uM sulforaphane for 48hr _vs_ vehicle | Sulforafan | PHH | 48 | 50 | 2619 | GSE20479 | 37.444 |
| MCF10A breast cell line + 15uM sulforaphane 24hr _vs_ vehicle control | Sulforafan | MCF10A | 24 | 15 | 2714 | GSE28813 | 21.824 |
| Primary human hepatocytes + 30uM oltipraz for 48hr _vs_ vehicle | Oltipraz | PHH | 48 | 30 | 2851 | GSE20479 | 29.187 |
| Hepatocytes of female donors treated 24hr with 600uM sulindac _vs_ 0uM | Sulindac | PHH | 24 | 600 | 1478 | TG-GATES | 11.854 |
| HepG2 hepatocellular carcinoma cell line 50uM Que treated for 24hr _vs_ DMSO control | Quercetin | HepG2 | 24 | 50 | 1849 | GSE28878 | 19.155 |
| HepG2 hepatocellular carcinoma cell line 50uM Que treated for 48hr _vs_ DMSO control | Quercetin | HepG2 | 48 | 50 | 3729 | GSE28878 | 26.481 |
| MCF10A breast cell line + Keap1 siRNA 24hr _vs_ control siRNA | *KEAP1* gene knockdown | MCF10A | NA | NA | 2267 | GSE28813 | 23.155 |
| A549 lung adenocarcinoma cells expressing NRF2 siRNA _vs_ non-targeting (NS) siRNA | *NFE2L2* gene knockdown | A549 | NA | NA | 3497 | GSE38332 | -40.398 |

PHH, primary human hepatocytes.

[1]It should be noted that A549 cells possess a homozygous mutation in Keap1 that results in increased NRF2 activity [33], explaining the dramatic effect on NRF2-regulated genes by knocking down *NFE2L2*.

## Building the experimental database

An annotated master database of gene expression profiles was created using BSCE, as previously described [11, 13, 22]. Briefly, BSCE contains over ~22,900 highly curated, publicly available, omic-scale studies from 15 species including ~140,000 lists of statistically filtered genes (as of June 2019). All available information for each human bioset (45,163 total) was downloaded and assembled into a database of experimental parameters. Each entry in the database contained the bioset name, GEO accession number (where applicable), analysis platform, and tissue (when available), and was then annotated for category of perturbant (i.e. chemical, gene, etc.), specific perturbant studied, and for chemicals, dose and time of exposure. This database was then populated with–log(p-values) from the Running Fisher algorithm (performed in the BSCE environment) to assess the correlation in gene expression changes between each bioset and the NRF2 biomarker. The Running Fisher test is a platform-independent, rank-based comparison that is otherwise similar to Gene Set Enrichment Analysis [32]. P-values were exported, converted to–log(p-values), and biosets with negative correlations were assigned negative values. Based on our past studies [11, 12], we considered–log(p-values) $\geq$ 4 to indicate NRF2 activation and $\leq$ -4 to indicate NRF2 suppression. In these past studies, a p-value cutoff <0.001 with a Benjamini-Hochberg multiple correction consistently resulted in a p-value cutoff of 1E-4 which gave balanced accuracies > 90% for biomarkers derived from human, mouse and rat transcript profiles. Only biosets examining the effects of individual chemicals (as opposed to more than one chemical) were evaluated in this study.

## Testing biomarker accuracy

To test for predictive accuracy, the biomarker was compared to gene expression biosets from chemically-treated HepG2 cells or primary human hepatocytes curated in the BSCE database. The biosets spanned exposures to 137 chemicals with known NRF2 activity based on ToxCast and Tox21 high-throughput NRF2 assay results. The comparisons were limited to HepG2 cells because this was the cell line in which the ToxCast and Tox21 assays were carried out, as well as primary hepatocytes to increase the number of chemicals with available biosets. True positive NRF2-activating chemicals were defined as those chemicals classified as NRF2 activators in both the ToxCast NRF2 (ATG_NRF2_ARE_CIS, Attagene Inc, Durham, NC) and the Tox21 NRF2 (Tox21_ARE_BLA_Agonist) assays (US. EPA, ToxCast & Tox21 Summary Files Released Dec. 2014, http://www.epa.gov/chemical-research/toxicity-forecaster-toxcasttm-data). True negatives were defined as those chemicals that were inactive in both assays. The biosets were filtered for exposure conditions that would be more likely for NRF2 activation, i.e., times between 8 and 48 h, and doses no lower than the HTS-determined $AC_{50}$ values (for positive chemicals) or between 10 and 400 uM (for negative chemicals). Thirty-five chemicals in 59 unique comparisons remained after using these filters. Comparisons were carried out using the Running Fisher algorithm. The biosets used to create the biomarker (essentially the training set) were excluded from the test. The values for predictive accuracy were calculated as follows: sensitivity (true positive rate) = TP/(TP+FN); specificity (true negative rate) = TN/(FP+TN); positive predictive value (PPV) = TP/(TP+FP); negative predictive value (NPV) = TN/(TN+FN); balanced accuracy = (sensitivity+specificity)/2. It should be noted that our true positive and true negative chemicals are based on HTS assay results carried out in HepG2 cells. There is evidence that the responses to perturbations will differ between cell types [34]. Thus, our true positive and true negative chemicals may be less applicable to cell types other than liver cells. However, as the biomarker was designed using multiple cell types, we carried out this and other analyses on both the HepG2/hepatocyte-filtered database and the entire human database.

## Ingenuity pathway analysis

The NRF2 biomarker genes were analyzed using the canonical pathway and upstream analysis functions of Ingenuity Pathway Analysis (IPA, Qiagen Bioinformatics, Redwood City, California). IPA calculates significance using a right-tailed Fisher's Exact test. The *p*-value is the probability of the overlap between the NRF2 biomarker gene list and the IPA pathway gene list. Significant reported pathways have q-values < 1E-3. Upstream analysis uses the number of differentially expressed genes to predict upstream regulators of the biomarker genes. A Z-score is applied in the predictions of upstream analysis. A Z-score of > |2| is considered significant. Multiple test correction on p-values was carried out using the Benjamini-Hochberg method.

## Identification of interactions between NRF2 and NRF2 biomarker genes using ChIP-Seq data

Analysis using human-based chromatin immunoprecipitation sequencing (ChIP-Seq) data was carried out to identify genes directly regulated by NRF2. Briefly, ChIP-seq data from experiments with activated NRF2 in human cells were obtained from Gene Expression Omnibus (3 biological replicates from a sulforaphane (SFN) treated human lymphoid cell line [GSM922966, GSM922967; GSM922968]; 2 biological replicates from SFN-treated human bronchial epithelial cell line BEAS-2B [GSM1968263; GSM1968264]; and 2 biological replicates from a human adenocarcinomic alveolar basal epithelial cell line A549 [GSM2423705;

GSM2423706]). Raw sequencing reads downloaded from NCBI Sequence Read Archive (SRA) were mapped to the human genome (hg19) using the BWA (Burrows-Wheeler Aligner) [35]. Replicates were combined and NRF2 binding peaks were determined by MACS version 2.0 [36]. Genomic coordinates derived from these datasets were updated to human genome build GRCh38/hg38 using LiftOver [https://genome.ucsc.edu/cgi-bin/hgLiftOver]. Coordinates were annotated to bi-directional promoter features with 10 Kb using the "ChIPpeakAnno" function in the R Bioconductor package (version 3.19.3) [37] based on the current UCSC TxDb database (2019-10-21). Additionally, we identified antioxidant responsive elements (AREs) based on a position weight matrix (PWM) defined by ChIP-seq binding data [38]. These regions were evaluated using the "matchPWM" function in the R BioStrings package (v2.46.0) [https://www.rdocumentation.org/packages/Biostrings/versions/2.40.2/topics/matchPWM]. Matches were based on a minimum of 80% relatedness. It should be noted that ARE binding in association with target genes was a post hoc analysis and served to support, but not develop, the composition of the biomarker (S1 File).

## Identifying modulators of the NRF2 pathway

To screen for chemicals that modulate NRF2, the biomarker was compared to each of the human biosets in the BSCE database. The p-values and directions of correlation were exported, p-values were converted to–log(p-values), and those with negative correlations were given negative values. Our master experimental database was then populated with the–log(p-values) for each bioset, which were classified as NRF2 activating (-log(p-value) $\geq$ 4), NRF2 suppressing (-log(p-value) $\leq$ -4) or having no effect on NRF2 (-log(p-value) between -4 and 4). Using this database, several strategies were implemented to identify chemicals that consistently activated NRF2. First, a Fisher's exact test was used to determine which chemicals were enriched in the group of NRF2 active biosets (p-value <0.05 after a Benjamini-Hochberg multiple test correction). For example, there are 239 biosets with "Tobacco Smoke" as the specific perturbant, 83 of which resulted in NRF2 activation and 155 of which did not, whereas only 1429 biosets of the 45,163 total biosets result in activation. (It should be noted that this analysis included all biosets in the database, including those examining the effects of chemicals.) Thus, NRF2 active biosets were significantly enriched for the perturbant "Tobacco Smoke." The same approach was taken with NRF2-suppressive biosets. This approach works well if there are a sufficient number ($\geq$ 3) of biosets for each specific chemical; however, it does not take dose or exposure time into account.

In another approach to characterize chemicals that activate NRF2, the maximum–log(p-value) for each specific chemical generated in HepG2 cells or primary human hepatocytes was determined, without accounting for dose or exposure time, and used to make an active or inactive call. The resulting list of active and inactive chemicals was then compared with the active and inactive chemicals from the ToxCast and Tox21 HTS assays described above. Perturbants that suppressed NRF2 could not be compared with the HTS assays because the assays were designed to measure NRF2 activation and, therefore, were considered inactive for activation in these comparisons.

Dose-response, time-course, and chemical-induced effects on datasets with listed oncogenic mutations were examined in a limited set of chemicals as proof of concept. The limited number and specific choices of chemicals was driven mainly by available data for those chemicals. Where possible, data from multiple studies were combined to construct a–log(p-value) vs. dose or time relationship. In these cases, all data for a perturbant were considered, and individual dose- or time-response curves were constructed from studies carried out in a single cell type.

## Validation experiments with the SRXN1-GFP reporter cell line

To investigate specific predictions based on the biomarker and differences between biomarker and HTS predictions, we employed a real-time, SRXN1-GFP fluorescence based NRF2-activation reporter system [39]. Thirty chemicals were screened in single/concentration-response format, over time, for their ability to induce expression of an SRXN1-GFP fusion protein in the human hepatoma HepG2 cell line (ATCC (clone HB8065)). The generation and qualification of the HepG2-BAC-GFP-SRXN1 reporter cell line has been previously described [39]. SRXN1-GFP cells were maintained and exposed in DMEM high glucose supplemented with 10% (v/v) Fetal Bovine Serum, 25 U/mL penicillin and 25 μg/mL streptomycin. During culturing of both the parental and reporter cell lines, mycoplasma was checked every 2 months using PCR-based testing. Three days prior to imaging, cells were seeded in Greiner black μ-clear 384 wells plates at 20,000 cells per well. Cytoplasmic accumulation of SRXN1-GFP levels and propidium iodide staining were monitored for 24 h using a Nikon TiE2000 confocal laser microscope (lasers: 408 nm, 488 nm and 561nm), equipped with an automated stage and perfect focus system. Prior to imaging at 20x magnification, HepG2 cells were loaded for 45 min with 100 ng/mL Hoechst$_{33342}$ to visualize the nuclei. The time interval was ~40 min for both SRXN1-GFP and propidium iodide (PI) staining. All chemicals came from the ToxCast chemical inventory (kindly provided by Dr. Ann Richard).

Single-cell SRXN1-GFP intensity levels were transformed by log(intensity + 0.001) to attain a more symmetrical distribution. GFP-positive cell counts (GFP_pos) were defined as the fraction of cells per treatment-concentration that were above the 0.75 quantile + 0.25* of the inter quartile range of the matching vehicle control single-cell population GFP levels. Concentration-response fits of the GFP positive fraction were modeled with the lm function in R as function of the concentration using piece wise natural splines with 3 degrees of freedom: lm (GFP_pos ~ ns(dose_uM, df = 3)). The point-of-departure was calculated as the concentration where the fit intersects the threshold of 0.1 fraction of GFP positive cells + the upper 90% confidence limit of the regression at each matching concentration to account for replicate variability. The PI positive cells (PI_pos) represent the fraction of cells with at least 4 pixels of propidium iodide segmentation overlapping the cell segmentation. The cell counts (cell_-count) represent the log fold-change in the number of cells compared to the matched vehicle controls and was defined by log10(cell count plate i/ vehicle control cell count_plate i).

## Results

### Building the NRF2 biomarker and assessment of predictive accuracy

The strategy for constructing the NRF2 biomarker, testing predictive accuracy, and screening for modulators is summarized in Fig 1. Our strategy for constructing the NRF2 biomarker was similar to our previous efforts at building biomarkers for other transcription factors in which we utilized archived microarray profiles generated from tissues or cells exposed to known chemical modulators as well as profiles generated from cells in which the gene encoding the transcription factor was either knocked out or knocked down (e.g., [11, 22, 23]). The present strategy utilized diverse profiles from human cells under conditions known to robustly activate NRF2 and included 24–48 hr exposures to the NRF2 activators sulforaphane (Gene Expression Omnibus (GEO) accession numbers: GSE20479 and GSE28813), oltipraz (GSE20479), sulindac (TG-GATES), and quercetin (GSE28878) (Table 1). Chemical activation of NRF2 was also compared to chemical-independent genetic activation by including a profile from siRNA knockdown of the NRF2 negative regulator Keap1 (GSE28813). Biomarker genes were selected based on a number of criteria. These included directionally consistent changes in expression

in most or all of the biosets in which NRF2 was chemically- or genetically-activated with no expression changes in the opposite direction. A genetic filter was implemented resulting in only genes that exhibit opposite directional changes after NRF2 was knocked down by siRNA (GSE38332). Finally, genes were filtered for those with relatively robust average expression changes (fold change ≥ |1.5|) across the activating conditions.

The ability of the biomarker to accurately predict NRF2 activation was assessed by comparing the results of the biomarker predictions with those from two high-throughput screening (HTS) assays carried out as part of ToxCast and Tox21 screening programs. These assays measured the activity at antioxidant response element (ARE)-linked reporters in HepG2 cells. In order to increase the confidence of these classifications, only chemicals that were called positives or negatives in both assays were used. There were 36 chemicals screened in these assays that overlapped with microarray studies in HepG2 or human primary hepatocytes in our compendium. The biosets used to create the biomarker (essentially the training set) were excluded from the test. The biomarker was compared to the microarray profiles using the Running Fisher test. Each pair-wise correlation resulted in a p-value which was converted to–Log(p-value), and those with negative correlation were converted to negative numbers. The balanced accuracy for the biomarker was calculated based on the 59 unique comparisons of biosets with corresponding HTS data (Fig 2A). Information about the biosets used in the test are provided in S1 File. The biomarker correctly classified thirteen true positive and 42 true negative biosets, with 3 biosets as false positives (allyl alcohol, carbamazepine, coumarin) and one false negative (mefenamic acid), resulting in a sensitivity and specificity of 93%. The positive predictive value was 81%, and the negative predictive value was 98%. The expression of the biomarker genes after exposure to the incorrectly classified chemicals is shown in Fig 2B. In general, the expression patterns of the false positive chemicals were consistent with that of the biomarker itself (i.e., the positive biomarker genes were generally increased in expression whereas the negative biomarker genes were generally decreased in expression). The false negative chemical mefenamic acid approached significance (-Log(p-value) = 3.7) and exhibited a pattern in which all but one of the 11 altered genes were directionally consistent with the biomarker.

Attempts were made to evaluate shorter lists of NRF2-regulated genes for prediction. A six gene biomarker based on some of the most frequently cited NRF2 targets *HMOX1*, *NQO1*, *TXNRD1*, *SXRN1*, *GPX2*, and *AKR1B10* had a balanced accuracy of only 66% and a sensitivity of 36%. This analysis indicates that a biomarker with a more comprehensive set of NRF2-dependent genes is more predictive than shorter lists of hand-picked genes.

## Characterization of the NRF2 biomarker genes

The 143 NRF2 biomarker genes exhibited consistent expression across the chemical and *KEAP1* knockdown perturbations (Fig 3A). As expected, the biosets from chemical activator treated cells or cells in which *KEAP1* was knocked down exhibited statistically significant positive correlation to the biomarker (p-values ≤ $10^{-10}$). The bioset in which *NFE2L2* was knocked down exhibited significant negative correlation to the biomarker (p-value ≤ $10^{-40}$).

The 68 upregulated and 75 downregulated genes in the biomarker included many well-known NRF2 targets (e.g., *AKR1B10*, *AKR1C1*, *NQO1*, *TXNRD1*, *SRXN1*, *GCLC* and *GCLM*) [28, 38]. Similar to our mouse NRF2 biomarker [20], *HMOX1* did not pass the genetic filter. Many of these genes were annotated near genomic regions bound by human NRF2 in human cell lines. Of the 143 NRF2 biomarker genes, 38 (26.6%) of these were associated with the NRF2-bound ChIP-Seq loci near gene promoter regions (within 10 Kb) in at least one dataset derived from cell lines treated with NRF2-activating isothiocyanate, SFN, or have

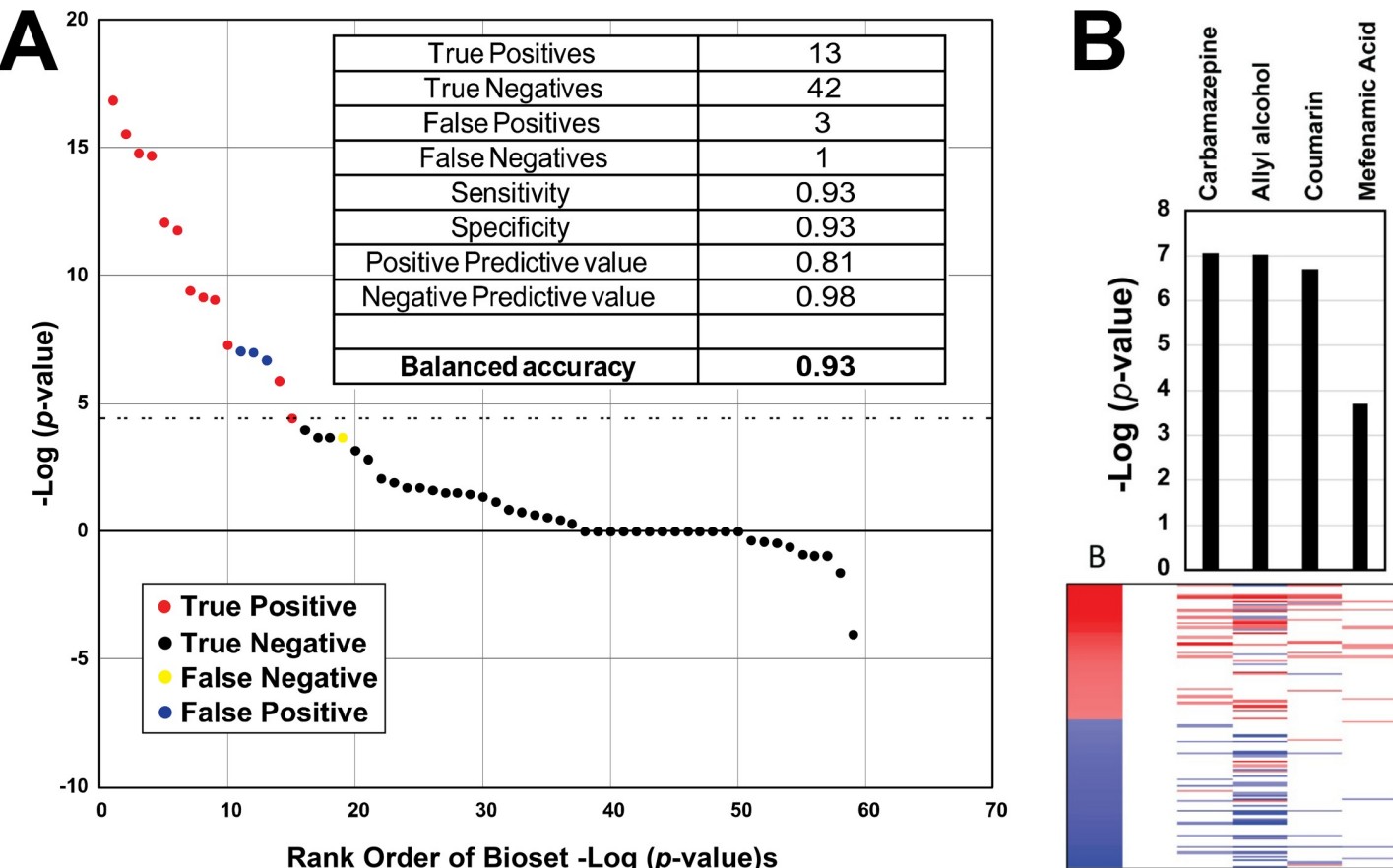

**Fig 2. Predictive accuracy of the NRF2 biomarker.** A. Predictions using the NRF2 biomarker were compared to those from the HTS NRF2 activation assays. The dashed line represents the cutoff value of four for significant biomarker activation. Those chemicals that were active (red) or inactive (black) in the HTS assays are shown. Three false positives (blue) and one false negative (yellow) are indicated. Inset, binary classifier calculations based on NRF2 HTS assay results. B. Expression changes of the biomarker genes for the three false positives and one false negative. (Top) The bar graph shows the -Log(p-values) of the comparisons between the biomarker and the indicated chemical. (Bottom) The heatmap shows the expression of the biomarker genes.

constitutively active NRF2 [33, 38, 40]. This is in comparison to the background rate of 13.7%, where a total of 3553 genes contained NRF2-ChIP bound regions within 10 Kb of the TSS of the approximately 26,000 identified transcribed genes in the human genome (hg38). Therefore, the NRF2 biomarker genes are more significantly represented with NRF2-bound regions (Fisher's Exact test, $p < 0.05$). Although these reference ChIP-seq loci were not derived from liver cells and could potentially miss regions that are liver-specific, we utilized these existing data from lung and lymphoblastoid cells to identify common NRF2-bound regions. Additionally, we identified 19 genes with evidence of NRF2-bound regions containing sequences that matched the ARE binding motif, supporting that some of these biomarker genes are directly regulated by NRF2 in a *cis*-acting manner (**S1 File**). Most of these genes (17 of 19) were activated in the biomarker, indicating that these genes are being transcriptionally upregulated by NRF2.

The NRF2 biomarker was evaluated for functional class enrichment via Ingenuity Pathway Analysis (IPA) (Fig 3B). The top canonical pathway identified as enriched with the biomarker genes was "NRF2-mediated oxidative stress". Other significantly enriched NRF2- linked pathways included "Glutathione biosynthesis" and "Thioredoxin pathway". NRF2 was identified as the top upstream regulating transcription factor that regulated the biomarker genes (Fig 3C).

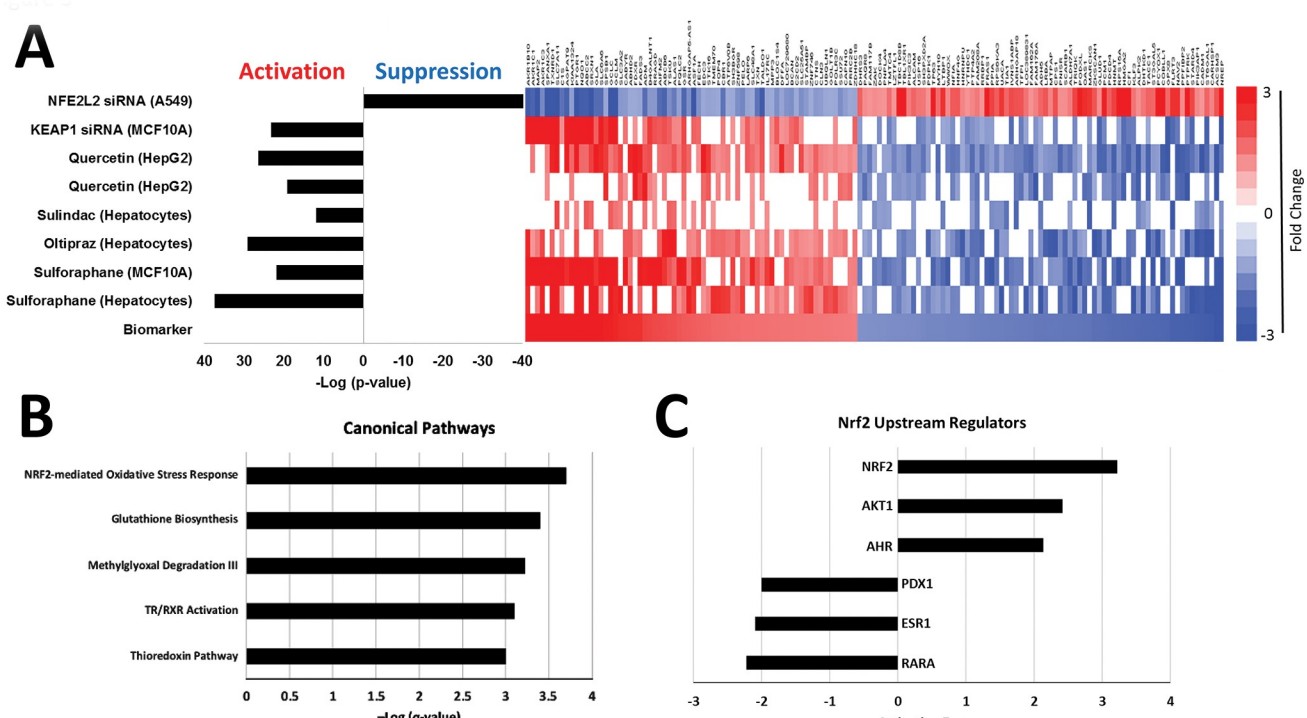

**Fig 3. Characterization of the human NRF2 biomarker.** A. Identification of NRF2 biomarker genes was based on consistent directional changes in gene expression resulting from exposure to NRF2 activators sulforaphane (GSE20479 and GSE28813), oltipraz (GSE20479), sulindac (TG-GATES), and quercetin (GSE28878), and knockdown of the NRF2 negative regulator Keap1 (GSE28813). Changes in gene expression were required to be in the opposing direction in cells in which *NFE2L2* (NRF2) expression was knocked down (GSE38332). The bars on the left represent the–Log (p-values) for the Running Fisher comparison test between the biomarker and the individual biosets used to construct it. The heatmap on the right depicts gene expression changes for the 143 genes in the biomarker across the individual biosets. B. Canonical pathway analysis of the biomarker genes. Multiple test correction on p-values derived from the right-tailed Fisher's exact tests was carried out using the Benjamini-Hochberg method. The -Log(q-value)s are shown. C. Potential upstream regulators of NRF2 biomarker genes (activation z-scores > |2|). Only upstream regulators with q-values < 0.05 are shown. Both analyses were conducted using Ingenuity Pathway Analysis software.

AKT and AhR were also significant upstream activators, both of which are upstream regulators of NRF2 [30, 41]. RARA, ESR1 and PDX1 were classified as upstream regulators that inhibit expression of the NRF2 biomarker genes. RARA (retinoic acid receptor alpha) and ESR1 (estrogen receptor alpha) signaling have previously been identified as inhibitory to NRF2 activation [42, 43]. *PDX1* (pancreatic and duodenal homeobox 1) is a diabetes susceptibility gene involved in pancreas development and regulates mitochondrial DNA transcription in pancreatic β-cells [44]. No direct interaction with NRF2 signaling has yet been identified for PDX1, however, indirect links between insulin signaling, PI3K, and NRF2 do exist [45].

## Screening for chemicals that modulate NRF2 in a gene expression compendium

The NRF2 biomarker was used to query the entire BSCE chemical database of ~9840 biosets, which examined the effects of ~2260 chemicals. Fig 4A shows the expression of the genes in the NRF2 biomarker across the biosets ranked by Running Fisher test significance. On the far left are those chemical comparisons that exhibited the greatest significant positive correlations to the biomarker. Some of the most significant chemicals included those used to create the biomarker as well as those that are well known to activate NRF2 (discussed below). These chemicals induced a pattern of expression of the biomarker genes markedly similar to that of the

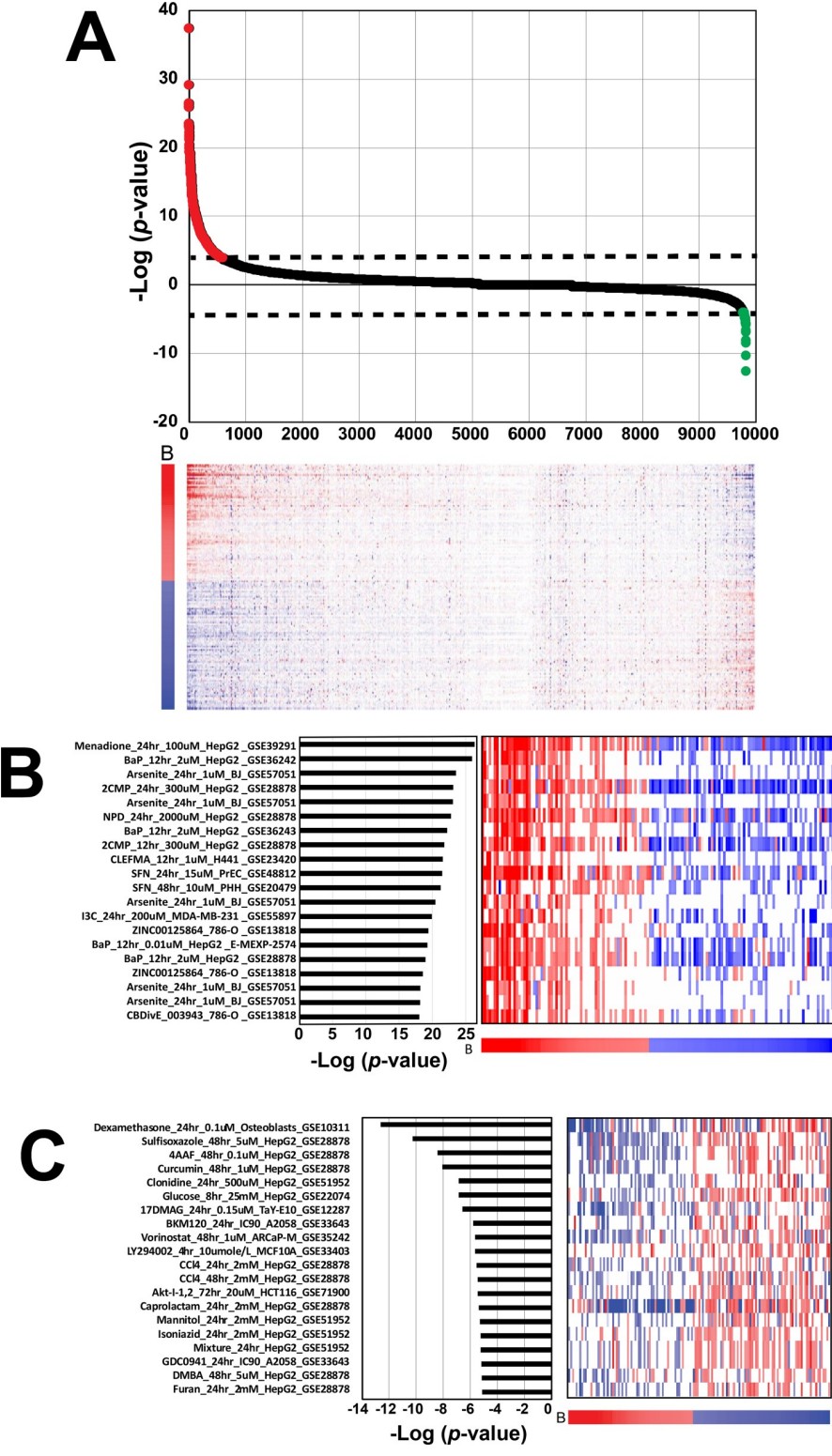

**Fig 4. NRF2 activity across biosets from chemically treated human cells.** A. (Top) Biosets derived from microarray comparisons of human cells exposed to chemicals were rank ordered based on their correlation to the biomarker using the -Log(p-value) of the Running Fisher test. Biosets with positive correlation (red) to the biomarker are on the left and biosets with negative correlation (green) to the biomarker are on the right. The dashed lines denote the cutoff p-value = $10^{-4}$. (Bottom) The heat map shows the expression of genes in the biomarker across the biosets. B, biomarker

genes. The numbers refer to rank-ordered bioset number. B. Top 20 chemical biosets that activate NRF2. Right, heatmap depicting the gene expression changes for the 143 genes in the NRF2 biomarker in each bioset. The biomarker gene expression changes are represented across the bottom of the heatmap. Each bioset is represented by the chemical, time and concentration of exposure, cell line used, and annotated study. One study did not have all information available (GSE13818). Abbreviations: BaP, benzo[a]pyrene; 2CMP, 2-(Chloromethyl) pyridine hydrochloride; NPD, 4-Nitro-o-phenylenediamine; CLEFMA, 4-[3,5-bis(2-chlorobenzylidene-4-oxo-piperidine-1-yl)-4-oxo-2-butenoic acid]; SFN, sulforafan; I3C, indole-3-carbinol. C. Top 20 chemical biosets that suppress NRF2. Abbreviations: 4AAF, 4-acetylaminofluorene; 17DMAG, 17-(dimethylaminoethylamino) 17-demethoxygeldanamycin; CCl4, carbon tetrachloride; Mixture, a mixture of liver toxicants; DMBA, 7,12-Dimethylbenz[a]anthracene.

biomarker itself. On Fig 4A, far right, a much smaller number of chemicals exhibited significant negative correlations to the biomarker. These chemicals induced a pattern of gene expression that was opposite to that of the biomarker genes, similar to that found when the *NFE2L2* gene was knocked down by RNAi (see above). These exposure conditions are thus thought to suppress the activity of NRF2.

There were 260 unique chemicals that exhibited NRF2 activation. The top 20 chemical biosets with the greatest correlation to the biomarker excluding the chemical comparisons used to construct the biomarker, included well-known NRF2 activators (benzo[a]pyrene, sodium arsenite, indole-3-carbinol, menadione, and sulforafan) (Fig 4B). Far fewer chemicals suppressed than activated NRF2 signaling. There were 43 chemicals that exhibited suppression of NRF2. The top 20 chemical biosets with the most significant negative correlation to the biomarker are shown in Fig 4C. The chemicals included carbon tetrachloride, curcumin, and dexamethasone. Approximately half of all the activating chemicals were from experiments in cell types other than HepG2 cells or primary hepatocytes, providing evidence that the biomarker has predictive ability outside the context of liver cells (S1 File).

The microarray compendium includes multiple comparisons assessing the same chemical but in different cell lines at different doses and times of exposure carried out by different labs. To help organize the predictions, a Fisher's Exact test was used to determine which chemicals were overrepresented in biosets with active NRF2 scores. Of the chemicals examined and which had a minimum of 3 bioset comparisons, 38 were significantly enriched ($p < 0.05$, with Benjamini-Hochberg correction) for NRF2 activation (Table 2). The most highly enriched chemicals included tobacco smoke, benzo[a]pyrene, sulforafan, quercetin, mitomycin C, and sodium arsenite, all of which are known NRF2 activators or generate reactive oxygen species [46]. Also included in the enriched chemicals were indole-3-carbinol, dithiothreitol (DTT) and a naturally occurring steroid compound, withaferin A.

There are several factors to consider when interpreting these results. First, experimental conditions other than the specific chemical (i.e. dose, time, cell line/type, etc.) are not included in this analysis, all of which can play a significant role in determining transcriptional responses. Second, the number of biosets a specific chemical is associated with can impact its significance in this test. For example, if a chemical is associated with only three biosets, all three must be positive for NRF2 activity for the chemical to be significantly enriched. Therefore, this strategy is helpful in identifying chemicals with numerous biosets that activate NRF2 or chemicals that consistently activate NRF2 in smaller numbers of comparisons.

## Dose-, time-, and mutation-dependent NRF2 modulation

Dose- and time-dependent NRF2 activation relationships were examined in a few select cases as proof of concept that the biomarker could be used to uncover characteristics of NRF2 activation. Biosets from HepG2 cells exposed to the polycyclic aromatic hydrocarbon benzo[a]pyrene, a prototypical AhR agonist and NRF2 activator, at 2 μM from four separate studies

**Table 2. Chemicals enriched in NRF2 active biosets.**

| Perturbant | Total Biosets | Active Biosets | Inactive Biosets | Fisher's Exact Test P-value (Benjamini-Hochberg Corrected) |
|---|---|---|---|---|
| Tobacco smoke | 239 | 84 | 155 | 1.39E-05 |
| Benzo(A)Pyrene | 27 | 15 | 12 | 2.78E-05 |
| Sulforafan | 10 | 10 | 0 | 4.17E-05 |
| Quercetin | 20 | 12 | 8 | 5.56E-05 |
| Mitomycin | 10 | 9 | 1 | 6.95E-05 |
| Smoke | 29 | 11 | 18 | 9.72E-05 |
| Sodium Arsenite | 11 | 7 | 4 | 1.11E-04 |
| Indole-3-Carbinol | 6 | 5 | 1 | 1.39E-04 |
| Dithiothreitol | 7 | 5 | 2 | 1.53E-04 |
| Withaferin A | 4 | 4 | 0 | 1.67E-04 |
| Mln4924 | 8 | 5 | 3 | 1.81E-04 |
| Azathioprine | 20 | 7 | 13 | 1.94E-04 |
| ((1S,2S,4R)-4-(4-((1S)-2,3-Dihydro-1H-Inden-1-Ylamino)-7H-Pyrrolo(2,3-D) Pyrimidin-7-Yl)-2-Hydroxycyclopentyl)Methyl Sulphamate | 5 | 4 | 1 | 2.08E-04 |
| Arsenic Trioxide | 5 | 4 | 1 | 2.22E-04 |
| Siomycin A | 5 | 4 | 1 | 2.36E-04 |
| Phenobarbital | 16 | 6 | 10 | 2.64E-04 |
| Nitric Oxide | 11 | 5 | 6 | 2.78E-04 |
| Oxyquinoline | 6 | 4 | 2 | 2.92E-04 |
| Decitabine | 146 | 16 | 130 | 3.06E-04 |
| Tetrachlorodibenzodioxin | 19 | 6 | 13 | 3.20E-04 |
| 4-(Acetoxymethylnitrosamino)-1-(3-Pyridyl)-1-Butanone | 3 | 3 | 0 | 3.33E-04 |
| Cresidine | 3 | 3 | 0 | 3.47E-04 |
| Menadione | 3 | 3 | 0 | 3.61E-04 |
| Phenylenediamines | 3 | 3 | 0 | 3.75E-04 |
| Pyridine Hydrochloride | 3 | 3 | 0 | 3.89E-04 |
| Rosemary Oil | 3 | 3 | 0 | 4.03E-04 |
| Tert-Butylhydroperoxide | 7 | 4 | 3 | 4.45E-04 |
| Azacitidine | 24 | 6 | 18 | 4.58E-04 |
| Oxidized-L-Alpha-1-Palmitoyl-2-Arachidonoyl-Sn-Glycero-3-Phosphorylcholine | 4 | 3 | 1 | 4.86E-04 |
| Vitamin K 3 | 10 | 4 | 6 | 5.14E-04 |
| Risperidone | 5 | 3 | 2 | 5.28E-04 |
| Atorvastatin | 12 | 4 | 8 | 5.42E-04 |
| Propylthiouracil | 12 | 4 | 8 | 5.56E-04 |
| 17b-estradiol | 174 | 15 | 159 | 5.70E-04 |
| Cisplatin | 157 | 14 | 143 | 5.83E-04 |
| Aflatoxin B1 | 13 | 4 | 9 | 5.97E-04 |
| WY 14,643 | 6 | 3 | 3 | 6.25E-04 |

The number of total biosets for each chemical were counted, as were the number of biosets in which NRF2 was activated or suppressed. A Fisher's exact test was used to determine which chemicals were significantly enriched in the group of biosets classified as NRF2 active.

(GSE28878, GSE36242, GSE36243, GSE40117) were examined in a time-course design (Fig 5A). Surprisingly, there was a strong negative, linear correlation between time of exposure (12 to 72 h) and –log(p-value) ($R^2$ = 0.904), indicating that NRF2 activation is decreasing with time of exposure beyond 12 hours, likely due to sequestration of reactive metabolites of BaP over time. Consistent with this, DNA adduct formation in HepG2 cells by BaP peaks at 12 hours

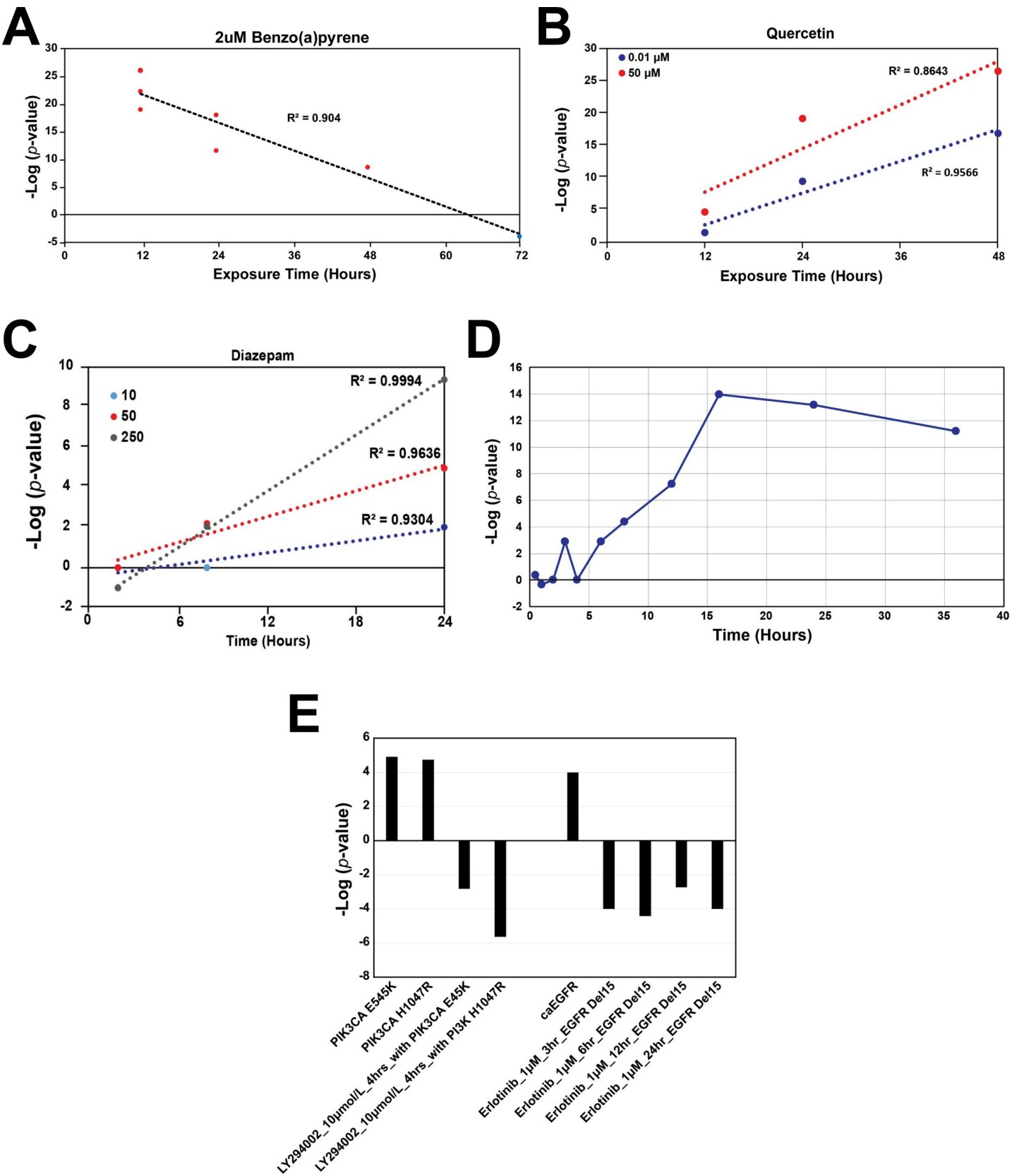

**Fig 5. Examples of dose-, time-, and mutation-dependent modulation of NRF2.** A. Response to 2 μM benzo[*a*]pyrene from 12 to 72 hours in HepG2 cells. Red circles indicate NRF2 activation, blue circles indicate inactivity. Data from GSE28878, GSE36242, GSE36243, GSE40117. B. Exposure of HepG2 cells to 0.01 μM and 50 μM quercetin from 12 to 48 hours. Data from E-MEXP-2574, GSE28878. C. Exposure of human primary hepatocytes to 10, 50, and 250 μM diazepam from 2 to 24 hours. Data from the TG-GATES study. D. Exposure of lung fibroblasts to dithiothreitol (2.5 mM) at different time points. Data from GSE4301. E. Examples of chemicals suppressing the activation of NRF2 in cancer cells expressing activated PI3K and EGFR. The activating mutations in PI3K3CA (E545K and H1047R) and overexpression of EGFR in the presence of EGF (caEGFR) lead to activation of NRF2 compared to wild-type cells. HCC827 with EGFR Del15 cells possess an amplified *EGFR* allele with an activating in frame deletion of 15 nucleotides in exon 19. Treatment of the cells with inhibitors for PI3K (LY294002) or EGFR (erlotinib) in the indicated cells suppresses background NRF2 activation.

compared to exposures of 6 and 18h [47]. In contrast, exposure to both 0.01 μM and 50 μM quercetin for 12–48 h in HepG2 cells from two studies (E-MEXP-2574, GSE28878) demonstrated strong positive, linear correlations ($R^2$ = 0.957 and 0.846, respectively) (Fig 5B). Diazepam exposure at 10, 50, and 250 μM for 2, 8 and 24 h in primary human hepatocytes (from the TG-GATES study) displayed strong positive, linear correlations ($R^2$ = 0.930, 0.964, and 0.999, respectively) (Fig 5C), demonstrating that prolonged diazepam exposures are required to activate NRF2. The time-dependent activation of NRF2 in lung fibroblasts by the known NRF2 activator, dithiothreitol, from GSE4301 was examined (Fig 5D). NRF2 was maximally activated by DTT at 16 hours.

Two chemicals were found to suppress NRF2 under conditions in which NRF2 exhibited constitutively higher background than wild-type cells. Two activating mutations in the phosphatidylinositol-4,5-bisphosphate 3-kinase, catalytic subunit alpha gene (*PI3K3CA*) (E545K and H1047R) (GSE33403) and overexpression of the epidermal growth factor receptor (*EGFR*) in the presence of EGF (caEGFR) (GSE3542) led to NRF2 activation compared to wild-type cells (Fig 5E). Treatment of cells carrying the activating mutations in PI3KCA by the PI3K inhibitor LY294002 led to suppression of NRF2. HCC827 cells possessing an amplified *EGFR* allele with an activating in-frame deletion of 15 nucleotides in exon 19 were treated with the EGFR inhibitor erlotinib, resulted in suppression of the background NRF2 activation.

## Comparison between biomarker predictions and HTS studies: Validation of predictions with a real-time NRF2 activation assay

Ninety chemicals screened in the ToxCast and/or Tox21 NRF2-activation assays were found in the microarray compendium. Prediction of NRF2 activity agreed for 65 (72%) of the chemicals, including seventeen actives and 48 inactives between the two HTS assays and using the biomarker (Fig 6). The biomarker identified thirteen actives that the HTS assays did not, and the HTS assays identified twelve actives that the biomarker did not.

Thirty chemicals were selected for follow-up screening in the SRXN1-GFP assay. The SRXN1-GFP assay identifies chemicals that activate NRF2 detected by increased expression of the GFP reporter under control of the human *SRXN1* promoter in HepG2 cells [39]. The chemicals were selected in part from the lists of NRF2 activators that differed between HTS assays and biomarker predictions (Table 3). Cells were treated at multiple concentrations and examined for GFP accumulation over a 24-h period. Thirteen chemicals (2-nitrofluorene, acetaminophen, allyl alcohol, atorvastatin, benzoin, carbamazepine, coumarin, diethylstilbestrol, disulfiram, mefenamic acid, resorcinol, tetracycline, tolbutamide) were selected that had a positive response based on the biomarker approach. Of these, 7 (2-nitrofluorene, allyl alcohol, carbamazepine, diethylstilbestrol, mefenamic acid, resorcinol, tetracycline) induced NRF2 activity at least at one concentration (Fig 7). The two positive controls quercetin and sulindac that were used to build the biomarker were positive in the assay. Of the remaining six inactive compounds, four (acetaminophen, benzoin, coumarin, tolbutamide) were tested in the microarray studies at concentrations exceeding what could be achieved in the GFP reporter assay.

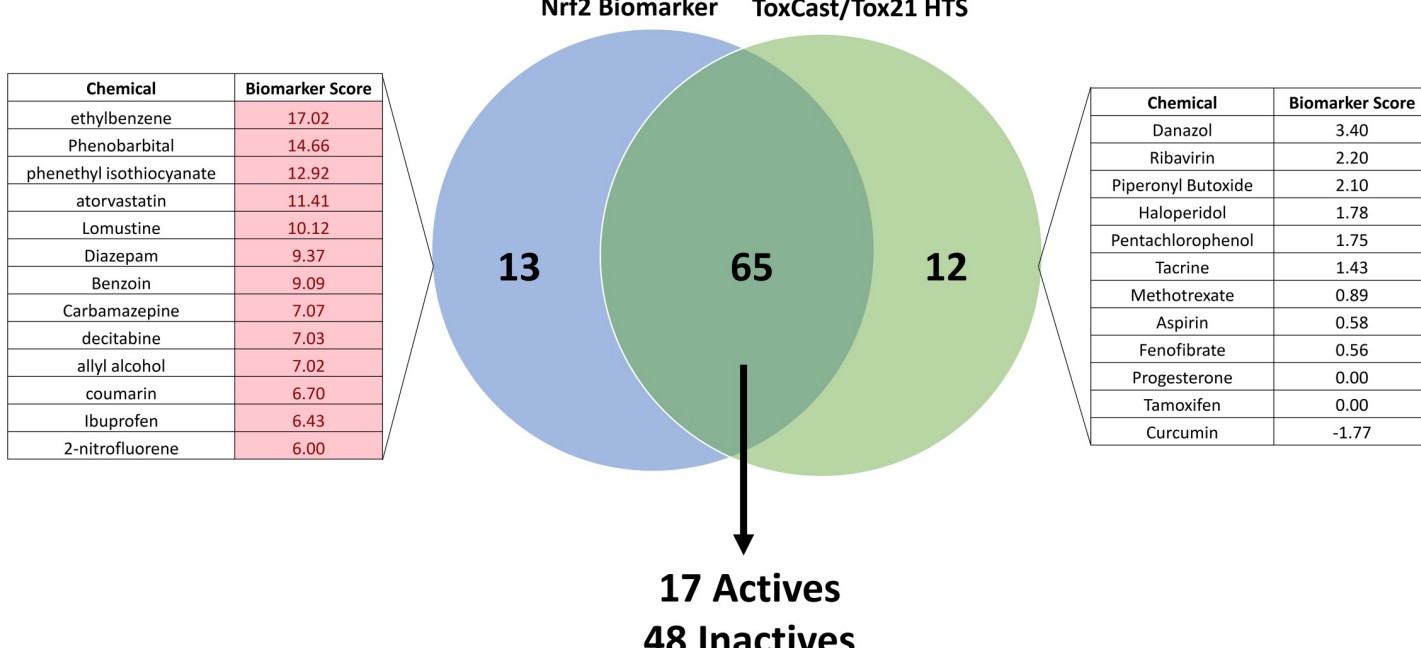

**Fig 6. Biomarker prediction agreement with HTS NRF2 assay results.** Maximum biomarker scores for each chemical were extracted from the database of HepG2 and hepatocyte experiments, and activity calls based on these scores were compared with activity determinations from ToxCast and Tox21 NRF2 HTS assays. Chemicals examined at a concentration above the HTS-determined cytotoxicity threshold for that chemical were excluded from the analysis. Lists represent those chemicals classified as NRF2-active by the biomarker only (left) or by HTS only (right). Biomarker score values shaded in red indicate results that are ≥ -Log(p-value) = 4 for activation of NRF2 signaling.

Thus, it is likely that NRF2 was not activated by these chemicals because of insufficient concentration (Table 3). Of the fourteen chemicals predicted to be inactive using the biomarker approach, nine (fenofibrate, progesterone, propiconazole, simazine, simvastatin, sulfisoxazole, tamoxifen, triclosan, valproic acid) were not active in the reporter assay. Four (danazol, hydroquinone, indomethacin, pentachlorophenol) of the remaining five chemicals predicted to be inactive using the biomarker were active in the reporter assay but were tested in the microarray studies at concentrations under most of the concentrations used in the reporter assay. Thus, these chemicals appear to be active at the concentrations tested in the reporter assays but not at the lower concentrations used in the microarray studies. Using the designations in Table 3 for the compounds studied, we determined predictive accuracy. The sensitivity, specificity, positive predictive value, and negative predictive value were 0.75, 0.82, 0.75 and 0.82, respectively. The balanced accuracy was 0.78.

In addition, one of the test chemicals, the PI3K inhibitor LY294002, acted as a NRF2 suppressor in the context of a constitutively active PIK3CA in the human breast cancer cell line MCF10A when tested at 20 μM (see above). We predicted that this compound would suppress background levels of NRF2 activation, but instead discovered it to be an activator in the reporter assay, albeit at concentrations that concurrently caused cytotoxicity.

## Discussion

High-throughput transcriptomic (HTTr) technologies have the potential to identify molecular targets in *in vitro* screens of environmental chemicals. In the present study, we used a computational approach to identify modulators of NRF2, the major regulator of cellular responses to oxidative stress. Using microarray data from chemical exposure and genetic modulation

**Table 3. Comparison of chemical effects using the biomarker and the SRXN1-GFP assay.**

| Chemical | CAS Number | Biomarker maximum (-log(p-value)) | Highest no-effect concentration for NRF2 activation in microarray experiments | Range of NRF2-active concentrations in microarray studies | Maximum Tested Concentration in SRXN1-GFP Assay (uM) | Minimum Tested Concentration SRXN1-GFP in Assay (uM) | Summary of SRXN1-GFP assay | Prediction of SRXN1-GFP results by microarray |
|---|---|---|---|---|---|---|---|---|
| 2-Nitrofluorene | 607-57-8 | 6 | | 32–18 uM | 200 | 0.1 | Active | TP |
| Acetaminophen | 103-90-2 | 6.522879 | | 10–1 mM | 200 | 0.1 | Inactive | 2 |
| Allyl alcohol | 107-18-6 | 7.022276 | | 70uM | 200 | 0.1 | Active | TP |
| Atorvastatin | 134523-00-5 | 11.40894 | | 40uM | 190 | 0.1 | Inactive | FP |
| Benzoin | 119-53-9 | 9.091515 | | 345uM | 200 | 0.1 | Inactive | 2 |
| Carbamazepine | 298-46-4 | 7.065502 | | 300uM | 200 | 0.1 | Active | TP |
| Coumaphos | 56-72-4 | 1.793174 | 250uM | | 200 | 0.1 | Active | FN |
| Coumarin | 91-64-5 | 6.69897 | | 300uM | 200 | 0.1 | Inactive | 2 |
| Danazol | 17230-88-5 | 3.39794 | 35uM | | 200 | 0.1 | Active | 1 |
| Diethylstilbestrol | 56-53-1 | 8.585027 | | 5uM | 200 | 0.1 | Active | TP |
| Disulfiram | 97-77-8 | 5.823909 | | 60uM | 200 | 0.1 | Inactive | FP |
| Fenofibrate | 49562-28-9 | 0.556737 | 30uM | | 200 | 0.1 | Inactive | TN |
| Hydroquinone | 123-31-9 | 1.549751 | 150uM | | 200 | 0.1 | Active | 1 |
| Indomethacin | 53-86-1 | 3.69897 | 200uM | | 200 | 0.1 | Active | FN |
| LY294002 | 154447-36-6 | -5.638 | 20uM | | 200 | 0.1 | Active | NU |
| Mefenamic acid | 61-68-7 | 4 | | 150uM | 200 | 0.1 | Active | 1 |
| Pentachlorophenol | 87-86-5 | 1.752027 | 10uM | | 100 | 0.05 | Active | 1 |
| Progesterone | 57-83-0 | 0 | 6uM | | 200 | 0.1 | Inactive | TN |
| Propiconazole | 60207-90-1 | 1.847712 | 10uM | | 200 | 0.1 | Inactive | TN |
| Quercetin | 117-39-5 | 26.48149 | | 10nM, 50uM | 200 | 0.1 | Active | PC |
| Resorcinol | 108-46-3 | 10.40894 | | 2mM | 200 | 0.1 | Active | TP |
| Simazine | 122-34-9 | -0.96981 | 50uM | | 200 | 0.1 | Inactive | TN |
| Simvastatin | 79902-63-9 | 0.407601 | 30uM | | 200 | 0.1 | Inactive | TN |
| Sulfisoxazole | 127-69-5 | -2.92082 | 5uM | | 200 | 0.1 | Inactive | TN |
| Sulindac | 38194-50-2 | 11.85387 | | 0.6, 3mM | 200 | 0.1 | Active | PC |
| Tamoxifen | 10540-29-1 | 0 | 25uM | | 100 | 0.05 | Inactive | TN |
| Tetracycline | 60-54-8 | 10.88606 | | 2mM | 200 | 0.1 | Active | TP |
| Tolbutamide | 64-77-7 | 9.744727 | | 2.1mM | 200 | 0.1 | Inactive | 2 |
| Triclosan | 3380-34-5 | -0.61834 | 22uM | | 200 | 0.1 | Inactive | TN |
| Valproic acid | 99-66-1 | 1.171985 | 5mM | | 200 | 0.1 | Inactive | TN |

Biomarker maximum (-log(p-value)): for any chemical represented by more than one bioset in the compendium, the top -Log(p-value) was selected. The range of concentrations assessed using the biomarker were from experiments in hepatocytes or hepatocyte-derived cell lines. In the column "Prediction of SRXN1-GFP results by microarray": PC, positive control; 1 = concentration tested in the microarray studies was under that tested in the GFP assay; 2 = concentration tested in the microarray study exceeded that tested in the GFP assay; NU = not used in prediction. LY294002 was tested to determine if this compound suppresses NRF2 activity (described in Fig 5E).

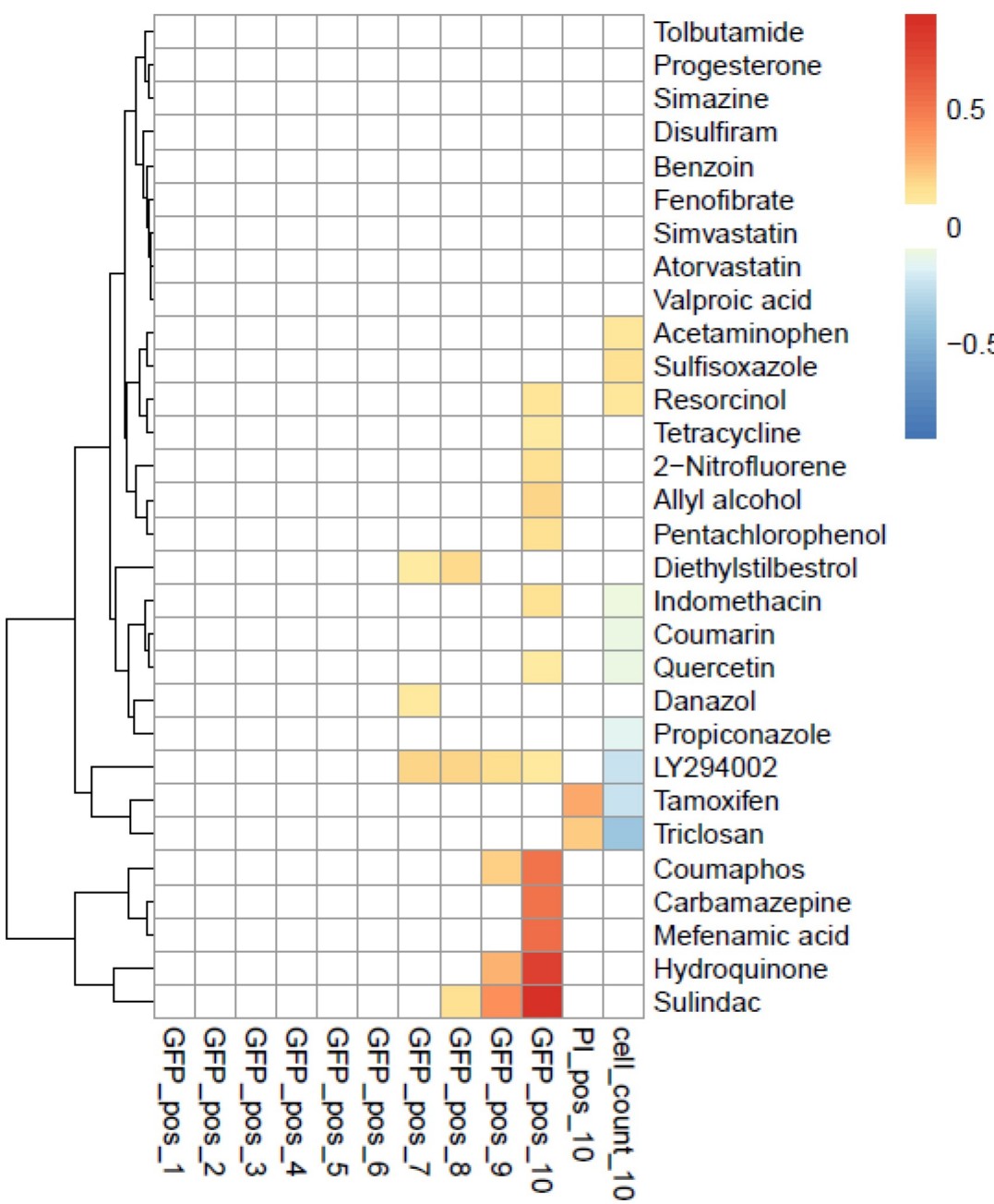

**Fig 7. NRF2 activity of selected chemicals in HepG2 cells.** Thirty chemicals tested for confirmation in the SRXN1-GFP assay. Cells were treated at multiple concentrations and examined for GFP accumulation over a 24-h period (GFP_pos_1–10) for concentrations 1 to 10. The last two columns, PI_pos_10 and cell_count_10, are the Propidium Iodide positive fraction of cells (marker for necrosis/ dead cells) and cell count compared to the negative control, respectively.

conditions known to either activate or suppress NRF2 activity, we identified 143 biomarker genes with an expression pattern consistent with NRF2 activation. When used in conjunction with a pattern-matching approach, our methods could readily identify, in a large gene expression human cell line compendium, chemical exposure experiments known to modulate NRF2. Using classifications from two HTS assays of NRF2 activation carried out in HepG2 cells, we found that the approach was highly predictive in identifying NRF2 activators (balanced

accuracy of 93%). In an independent study of 29 chemicals, the predictions based on the NRF2 biomarker were independently validated for most of the chemicals in HepG2 cells encoding a NRF2-responsive GFP reporter (78% balanced accuracy). Our methods were used to perform a virtual screen of chemicals in a human gene expression compendium of ~2380 chemicals. We found that 288 chemicals activated NRF2 including several chemicals that were not also identified in the NRF2 HTS assays. We identified 53 chemicals that appeared to suppress NRF2, because exposure led to an expression pattern opposite that of the biomarker itself. The results indicate that our approach can be reliably used as a Tier 1 screen in the context of a larger HTTr profiling effort, similar to those ongoing in the ToxCast screening program [48].

Determination of predictive accuracy required identification of a set of chemicals with known (positive or negative) activity for NRF2. In our microarray compendium there were relatively few chemicals that could be considered "true" positive or negative NRF2 activators, because many chemical activators reported in the literature have been investigated only at the level of transcriptional induction of one or a few NRF2 target genes, far less than the full-genome analysis in the present study. Microarray studies of dose-response and time-course relationships are likewise relatively rare. Thus, the number of chemicals that can be confidently classified as true activators within the human compendium were too few to perform meaning-ful balanced accuracy calculations. We therefore chose to leverage the ToxCast and Tox21 HTS assay data to classify chemicals as NRF2 activators. Only chemicals consistently active or inactive in both assays were used to determine accuracy. These selection criteria yielded 35 chemicals with consensus NRF2 HTS activity. Using these designations as the truth, the bal-anced accuracy of the biomarker was 93%. There were three false positives (allyl alcohol, carba-mazepine, coumarin) and one false negative (mefenamic acid). The microarray profiles of these compounds exhibited expression of biomarker genes that were consistent in direction and magnitude to the changes of the biomarker genes. Remarkably, carbamazepine was weakly active at 100 μM, and coumarin was active at 50 μM in earlier studies carried out in HepG2 cells [49, 50], indicating that despite the lack of activity in both HTS assays, these chemicals are likely NRF2 activators. Reclassification of these two chemicals as true positives, improves bal-anced accuracy to 96%. The excellent predictive accuracy is similar to those of other biomark-ers that we have characterized, including those that identify chemical modulators of xenobiotic-activated transcription factors in the mouse and rat liver [11–13, 16, 17, 19] and those that identify modulators of ER and AR in human cell lines [22, 23].

In addition to the chemicals discussed above, which had consistent NRF2-activating behav-ior in both HTS assays, we expanded our comparison to the larger set of chemicals in which there were differences between the two HTS assays. In general, our predictions of NRF2 activa-tion compare favorably with those made from ToxCast and Tox21 *in vitro* HTS assays. After filtering for biosets exposed at concentrations higher than the HTS-derived cytotoxicity threshold, predictions for 65 chemicals agreed between the biomarker and HTS approaches (Fig 6). There were thirteen chemicals that the biomarker identified as activating NRF2 that were not called hits in the HTS assays. This may be the result of cell type differences, because primary hepatocyte data was included in the biomarker-based approach. HTS assays identified twelve active chemicals classified as inactive by the biomarker. However, these microarray studies were not optimal for comparison to the more standard HTS data. The chemicals included five in which exposures were below the HTS-determined $AC_{50}$ (i.e., test concentra-tions insufficient to activate NRF2), four chemicals were tested using exposure durations that may not be optimal for NRF2 activation, including three that were exposed for 8 h or less, and one for 72 h. Two chemicals had no dose information available. Thus, the microarray data available for these chemicals was suboptimal. One notable difference in the chemicals identi-fied by HTS assays but not the biomarker was curcumin. Curcumin is a diarylheptanoid found

in turmeric and is well known to increase NRF2 signaling [51, 52]. HTS assays correctly identified curcumin as a hit, whereas the biomarker did not. Surprisingly, gene expression changes in HepG2 cells exposed to 1 μM for 12, 24, or 48 hours (GSE28878) resulted in significant NRF2 suppression at 48 hours, exactly opposite of that expected. Regarding the chemicals that were classified as activating NRF2 by the biomarker approach but not in the HTS assays, it is possible that the conditions used to assay NRF2 in the HTS assays was not optimal. Most of the chemicals were either examined at doses above 100 μM (eight chemicals) or at time points greater than 24 hours (three chemicals), or both in the microarray studies. Only allyl alcohol was examined under conditions comparable to the HTS assays (70 μM and at 24 hours). Differences in chemical metabolism may also contribute to differences between assays. Thus, the differences in the classifications between the HTS assays and the biomarker could be explained in part by differences in the concentrations and times of exposure. These considerations are important given that there are differences between chemicals in the exposure times that produce maximal activation of NRF2 (Fig 5).

Given some inconsistencies between our biomarker approach and the HTS assays, we examined 29 chemicals in HepG2 cells that possess a NRF2-reponsive reporter in which GFP is under control of the promoter from the *SXRN1* gene. We found that when dose is considered, most of the predictions based on the biomarker approach could be confirmed using the reporter system. In any toxicity assay, false negatives are a concern. Out of the thirteen chemicals that were positive using the biomarker approach, six chemicals tested inactive in the reporter assay. However, all but two of the chemicals (atorvastatin, benzoin) were active only at concentrations that could not be achieved using the reporter assay due to DMSO tolerance of the HepG2 cells coupled to stock solution concentrations. Excluding those chemicals in which sufficient test concentrations could not be achieved, the reporter assay was able to confirm 7 of the 9 chemicals that were identified as positive using the biomarker (Table 3). In summary, the SXRN1-GFP assay was able to predict the NRF2 activation status of most of the chemicals selected for rescreening (78% balanced accuracy).

We performed a virtual screen of the chemicals in the human microarray compendium and found many chemical-dose-time conditions that result in NRF2 activation. The biomarker correctly identified many known chemical NRF2 activators (e.g., sodium arsenite, phenylenediamines, azathioprine, dithiothreitol), as well as known environmental stressors that increase oxidative stress (tobacco smoke, nitric oxide). Furthermore, the procedures identified chemicals that interact with the aryl hydrocarbon receptor (benzo[a]pyrene, smoke) and whose CYP-mediated metabolism results in increased oxidative stress, activating NRF2 [41]. Importantly, the biomarker identified NRF2-activating chemicals in cell lines other than HepG2 or primary hepatocytes. This finding was not unexpected because the biomarker was built from data derived from multiple cell types. Although there are likely cell type-specific effects of NRF2 activation (e.g., [34, 53]), there are also core changes in gene expression that can be used as a marker for activation in a larger number of cell lines.

In addition to NRF2 activators, our method identified chemicals that suppress NRF2 activity. Forty-three chemicals elicited a pattern of gene expression with significant negative correlation to the biomarker, similar to the pattern observed with NRF2 knockdown. Although it is possible that fewer chemicals suppress than activate NRF2, it is likely that the experimental conditions across the compendium were not optimized to identify NRF2 suppressors. To effectively identify suppressors, an attempt to first stimulate NRF2 activity is needed, similar to running the "antagonist mode" in hormone receptor assays in which cells are pre-treated with a test compound before the addition of a known agonist. In fact, two compounds identified by the biomarker as NRF2 suppressors, the PI3K inhibitor LY294002 and the EGFR erlotinib, did so in the context of cells harboring oncogenic mutations that led to constitutive activation of

PI3K in MCF10A cells [54] or lung cancer cells expressing a constitutively active EGFR [55]. Examination of two constitutively active mutants in PI3KCA vs. wild-type cells or overexpression and activation of EGFR vs wild-type cells showed that NRF2 was activated supporting earlier work [30]. We tested the ability of the LY294002 compound to suppress NRF2 in our SRXN1-GFP assay, but instead we were surprised that NRF2 was activated. The activation may be an off-target effect since the NRF2-activating concentration of the chemical was much higher than that used to inhibit PI3K. Overall, while screening for NRF2 suppressors appears to be feasible using our approach, the microarray data in the compendium was not generated under conditions optimized to allow suppression of constitutively activated NRF2. In summary, the present study is to our knowledge the first to utilize gene expression data to identify potential human NRF2 modulators in a large screening study. Using a similar approach, our group has recently identified chemicals that activate NRF2 in the mouse liver and compared the activation to conditions in which a number of xenobiotic-activated transcription factors are chemically activated [20, 21, 56].

One focus of the ToxCast high-throughput screening program at EPA is to prioritize for further testing the vast number of chemicals used in industry with little toxicity information. Gene expression profiling represents a more global screening approach and an important complementary system to the current *in vitro* HTS assays that are at the forefront of the field. The historical challenge of low throughput of gene expression analysis, has been partially addressed through advances in technology. Another major challenge since the beginning of toxicogenomics is linking changes in gene expression to specific molecular events and toxicological outcomes. Our work here in the case of NRF2 and by extrapolation increases in oxidative stress, and in other publications examining endpoints important in endocrine disruption [22, 23] and DNA damage [24, 57], directly addresses this issue. Moving forward, biomarkers for activation of transcription factors can, to some extent, be developed using existing microarray data. However, there are many important targets of environmental chemicals that lack the chemical activator data required to construct predictive biomarkers. Cell-based experiments utilizing siRNA knockdown or Crispr-Cas9 mediated knockout of transcription factors of interest combined with exposure to chemical activators and gene expression profiling via current, high-throughput approaches have the potential to quickly fill these data gaps.

## Supporting information

**S1 File. Supporting information.** Contains 1) list of genes that comprise the NRF2 biomarker, 2) evidence from ChIP-Seq studies for direct interactions between biomarker genes and NRF2, and 3) biosets examined in this study.
(XLSX)

## Acknowledgments

We thank two anonymous scientists for critical review of the manuscript, Dr. Ann Richard for the ToxCast chemicals, Drs. James Flynn and Joe Delaney for assistance with BSCE, and Chuck Gaul and Molly Windsor for assistance in making the figures.

**Disclaimer:** The information in this document has been funded in part by the U.S. Environmental Protection Agency. It has been subjected to review by the Center for Computational Toxicology and Exposure and approved for publication. Approval does not signify that the contents reflect the views of the Agency, nor does mention of trade names or commercial products constitute endorsement or recommendation for use.

## Author Contributions

**Conceptualization:** John P. Rooney, J. Christopher Corton.

**Data curation:** John P. Rooney, Brian Chorley, Steven Wink, Douglas A. Bell, J. Christopher Corton.

**Formal analysis:** John P. Rooney, Brian Chorley, Steven Hiemstra, Steven Wink, Xuting Wang, Bob van de Water, J. Christopher Corton.

**Investigation:** John P. Rooney, Bob van de Water, J. Christopher Corton.

**Methodology:** John P. Rooney, Douglas A. Bell, J. Christopher Corton.

**Project administration:** J. Christopher Corton.

**Resources:** J. Christopher Corton.

**Supervision:** J. Christopher Corton.

**Visualization:** J. Christopher Corton.

**Writing – original draft:** John P. Rooney, J. Christopher Corton.

**Writing – review & editing:** John P. Rooney, Brian Chorley, Steven Hiemstra, Steven Wink, Xuting Wang, Douglas A. Bell, Bob van de Water, J. Christopher Corton.

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
