## [Decision Letter · Decision Letter 0]

1 May 2020

PONE-D-20-07833

Mining a human transcriptome database for chemical modulators of NRF2

PLOS ONE

Dear Dr. Corton,

Thank you for submitting your manuscript to PLOS ONE. After careful consideration, we feel that it has merit but does not fully meet PLOS ONE’s publication criteria as it currently stands. Therefore, we invite you to submit a revised version of the manuscript that addresses the points raised during the review process. particukar care should be given to addessing concerns of Reviewer 2, who has been more critical.

We would appreciate receiving your revised manuscript by Jun 14 2020 11:59PM. To enhance the reproducibility of your results, we recommend that if applicable you deposit your laboratory protocols in protocols.io, where a protocol can be assigned its own identifier (DOI) such that it can be cited independently in the future. For instructions see: http://journals.plos.org/plosone/s/submission-guidelines#loc-laboratory-protocols

We look forward to receiving your revised manuscript.

Kind regards,

Roberto Mantovani

Academic Editor

PLOS ONE

2. Please provide additional information about the HepG2 cell line used in this work, including source, history, culture conditions and any quality control testing procedures (authentication, characterisation, and mycoplasma testing). For more information, please see http://journals.plos.org/plosone/s/submission-guidelines#loc-cell-lines.

3. Please note that PLOS does not permit references to “data not shown.” Authors should provide the relevant data within the manuscript, the Supporting Information files, or in a public repository. If the data are not a core part of the research study being presented, we ask that authors remove any references to these data."

4. To comply with PLOS ONE submission guidelines, in your Methods section, please provide additional information regarding your statistical analyses. For more information on PLOS ONE's expectations for statistical reporting, please see https://journals.plos.org/plosone/s/submission-guidelines.#loc-statistical-reporting.

"The information in this document has been funded in part by the U.S. Environmental Protection Agency. This research was supported in part by a postdoctoral appointment to JPR to the Research Participation Program for the U.S. Environmental Protection Agency, Office of Research and Development, administered by the Oak Ridge Institute for Science and Education through an interagency agreement between the U.S. Department of Energy and EPA."

"The authors received no specific funding for this work."

7. Thank you for stating in your Funding Statement:

"The information in this document has been funded in part by the U.S. Environmental Protection Agency. This research was supported in part by a postdoctoral appointment to JPR to the Research Participation Program for the U.S. Environmental Protection Agency, Office of Research and Development, administered by the Oak Ridge Institute for Science and Education through an interagency agreement between the U.S. Department of Energy and EPA."

8. We note that you have included the phrase “data not shown” in your manuscript. Unfortunately, this does not meet our data sharing requirements. PLOS does not permit references to inaccessible data. We require that authors provide all relevant data within the paper, Supporting Information files, or in an acceptable, public repository. Please add a citation to support this phrase or upload the data that corresponds with these findings to a stable repository (such as Figshare or Dryad) and provide and URLs, DOIs, or accession numbers that may be used to access these data. Or, if the data are not a core part of the research being presented in your study, we ask that you remove the phrase that refers to these data.

Reviewers' comments:

Reviewer's Responses to Questions

**Comments to the Author**

1. Is the manuscript technically sound, and do the data support the conclusions?

Reviewer #1: Yes

Reviewer #2: Yes

2. Has the statistical analysis been performed appropriately and rigorously? 

Reviewer #1: I Don't Know

Reviewer #2: Yes

3. Have the authors made all data underlying the findings in their manuscript fully available?

Reviewer #1: Yes

Reviewer #2: Yes

4. Is the manuscript presented in an intelligible fashion and written in standard English?

Reviewer #1: Yes

Reviewer #2: Yes

5. Review Comments to the Author

Reviewer #1: Gene expression data:

Is it possible to make the expression data available for your biomarker gene set, or does PLOS One require the publication of this data?

A short explanation of the commercial database used here or how this database ensures the integration of the different gene expression data (normalization strategy, ...) would improve the transparency of the data analysis. May be the authors can add a short paragraph instead of citing an article. My answer to the question: "Has the statistical analysis been performed appropriately and rigorously? -> I don't know" is related to this part.

Characterization of the NRF2 biomarker genes (ChIP-seq analysis):

The authors analyzed and verified the final biomarker gene set in HEPG2 cells. To what extent are the used ChIP-seq data suitable to make statements about regulatory areas in HEPG2 cells? To what extent can direct (motif-defined) or indirect binding of NRF2 be expected?

How do the authors linked the experimental ChIP-seq regions to the biomarker gene set? May be a motif analysis of a "defined" promoter-region is more meaningful here. Perhaps this analysis was done already (see link: supplemental file 1 and line 390 in the manuscript) but may be the authors can add more details here.

In this paper the ChIP-seq analysis was done based on Ensembl database version 86 (Okt 2016). Why did the authors use this older version of the database in this actual paper?

In figure 4 the term 'bioset' is used in the legend and in the paper. In the paper the term 'biomarker gene set' is used before. Do these two terms refer to the same gene set? A minor point but I was just confused about this two terms. May be it is a frequently used term, which I just do not know.

Dose-, time-, and mutation-dependent NRF2 modulation:

The authors compare different chemicals screened in the ToxCast and/or Tox21 studies and relate these results to NRF2. First liver-cells are shown but also fibroblast and breast cancer cell line data are shown. The authors validate their findings using ARE-linked reporter system in HepG2 cells. The comparison with non-liver-related results is confusing. How comparable the findings between different cell types. Is there a possibility to estimate the completeness of this gene set?

-----

Figure 1: Biomarker construction and screening strategy.

-> In all other Figure description the authors use Figure X. Description of the figure.

May be the ":" is a typo.

short comment to Figure 6:

May be the used red color should be defined in the legend of this figure.

Reviewer #2: The manuscript describes a very reasonable approach to using chemical perturbation transcriptional signatures to identify chemicals altering the activity of the transcription factor NRF2. I have several comments that I would like to see addressed by the authors.

Procedure used to construct the list of differentially expressed genes used as NRF2 biomarkers - Relevant sections: Methods (162-177), Results (334-338):

- I think some key details are missing from the methods. In general, the level of detail and tone of the methods section seems more adequate to the results section. Specifically:

- can you be more specific, and provide the logical rules that were used to construct these gene-sets, e.g. fold change ≥ 1.5 and adjusted p-value < ... in at least N/M NRF2 activation data-sets, and fold change < ... and adjusted p-value < ... in ... NRF2 activation data-set(s)?

- how was gene differential expression calculated? Please state what method was used (e.g. limma) and if a linear model was used, how were the experimental factors modelled? And what multiple test correction method. This applies to the analysis of other differential expression data-sets

- can you please specify if all genes were probed or only a subset? (some older microarray platforms only probed a subset of genes)

- how different were these biomarker gene-sets from each other? E.g. can you show a reciprocal overlap matrix, and report their sizes..?

Performance assessment:

- I would appreciate a bit more detail on the performance for all ten biomarker gene-sets. You used ToxCast and Tox21 data to pick the best biomarker gene-set, therefore these data-sets were in a sense part of your training set. You don't have a completely separate test set, and so the performance may be a bit inflated. In general, it is a good practice to use cross-validation in this type of situation. However, if the performances of the ten biomarker gene-sets are all quite similar, I would not be too worried about this (also recognizing that you don't have a lot of ToxCast and Tox21 datasets, and so implementing cross-validation may be a bit tricky, the set is small, so things may get noisy). For the same reason, it would be useful to know how different were these biomarker gene-sets from each other (as commented above). You could use the ChIP-seq data to argue that the selection of the best biomarker set is robust (see comment below).

- Is this PPV really general? Usually the PPV can be different from sensitivity and specificity if the number of real positives is swamped by the number of real negatives (e.g. a test detecting a disorder that's present in 0.1% of the population). You have 14 real positives and 45 real negatives, does this reflect the general distribution in reality, can you comment on this?

ChIP-seq and PWM analysis:

- the percentage of signature gene with a ChIP-seq mark is impressive (68.5%). I would like a bit more details. For the annotation using ChIPpeakAnno, how far from a gene (transcription start or end) could a peak be? What fraction of genes had a peak before performing any intersection with the biomarker gene set? Can you show that this is much better as e.g. taking genes that were upregulated in at least one NRF2 activation data-set? Can you show that the top biomarker gene-set had also better ChIP-seq overlap (following up on the issue of potential inflation in performance)

- similar considerations for the PWM analysis

Ingenuity pathway analysis

- can you please use multiple-test adjusted p-values? (or if you already did, make that explicit)

- can you provide more details on how this analysis is performed "top upstream regulating transcription factor that regulated the biomarker genes"

Screening for chemicals that modulate NRF2

- can you explicitly comment on the anticipated false discovery rate? My back of the envelope calculation is: 9840 biosets * 1e-4 = 0.984 biosets, which is pretty stringent (this is very simplistic as it does not take into account correlations, which are probably abundant)

- for chemicals represents in multiple data-sets, how about using the median fold-change to collapse redundancy..?

Follow-up screening using the SRXN1-GFP assay

- in my mind, this experiment should be used as follows: from the results of 'Screening for chemicals that modulate NRF2', pick some that induce a positive effect, some that induce a negative effect, some that don't induce an effect; this selection should be unbiased, and it should exclude chemicals whose differential expression was used to construct and optimize the biomarker set (e.g. quercetin, sulindac, ...). Testing genes used to construct the biomarker gene-set would make sense only to validate this assay (e.g. it is known that quercetin activates NRF2). Doses should also be matched. Instead, it appears this experiment was designed to answer multiple questions (e.g. are negative results in the ToxCast and Tox21 used to select the best biomarker gene-set really negative, even when the biomarker gene-set suggests otherwise?), and it's harder to draw firmer conclusions; for instance, several of the chemicals use to construct and optimize the best biomarker gene-set were included.

- for these reasons, I am not supportive of this statement in the discussion << In an independent study

of 29 chemicals, the predictions based on the NRF2 biomarker were independently validated for most (81%) of the chemicals in HepG2 cells encoding a NRF2-responsive GFP reporter. >>

Discussion

- it is quite long-winded and should be shortened/summarized a bit

Other / General

- is it possible to have both effect size and p-value for biomarker signature correlations using Running Fisher..?

Table 1

- Is it possible to indicate the number of differentially expressed genes?

Figure 2

- figure 2A has resolution issues, please remake it

6. PLOS authors have the option to publish the peer review history of their article (what does this mean?). If published, this will include your full peer review and any attached files.

Reviewer #1: No

Reviewer #2: No

---

## [Author Response · Author response to Decision Letter 0]

12 Jun 2020

Response to reviewer comments for

Mining a human transcriptome database for chemical modulators of Nrf2

John P. Rooney1,2, Brian Chorley2, Steven Hiemstra3, Steven Wink3, Xuting Wang4, Douglas A. Bell4, Bob van de Water3, and J. Christopher Corton2,5

We thank the two reviewers for their suggestions that greatly improve the clarity and impact of the manuscript. Our responses to each comment are found below in bold.

5. Review Comments to the Author

Reviewer #1: Gene expression data:

Is it possible to make the expression data available for your biomarker gene set, or does PLOS One require the publication of this data?

We have provided the list of the biomarker genes and their associated fold-change values in Supplemental File 1. Also in this file, we provide a master table of the different microarray comparisons (biosets) examined in this study including the GEO or ArrayExpress numbers of the publicly available datasets. We hope this addresses your question.

A short explanation of the commercial database used here or how this database ensures the integration of the different gene expression data (normalization strategy, ...) would improve the transparency of the data analysis. May be the authors can add a short paragraph instead of citing an article. My answer to the question: "Has the statistical analysis been performed appropriately and rigorously? -> I don't know" is related to this part.

The reviewer makes an important point about the data analysis procedures that should be clear in the paper. We have added text in the first paragraph in the Methods section to describe the gene expression workflow used in BaseSpace Correlation Engine including the criteria for selecting significantly altered genes. It should be noted that our group has used this database for the last 5 years resulting in >20 publications. The data is rigorously quality controlled. In fact, BSCE notes that they reject ~30% of all studies, because they do not meet their quality control criteria. We have great confidence that the methods that are used are rigorous and allow for appropriate comparisons between gene lists.

Characterization of the NRF2 biomarker genes (ChIP-seq analysis):

The authors analyzed and verified the final biomarker gene set in HEPG2 cells. To what extent are the used ChIP-seq data suitable to make statements about regulatory areas in HEPG2 cells? To what extent can direct (motif-defined) or indirect binding of NRF2 be expected?

There are no available NRF2 ChIP datasets that have been performed in NRF2 activated human hepatocyte or liver samples, therefore we had to rely on the limited published datasets that were currently available. Our previous studies in LCL (Chorley et al. NAR 2012) demonstrated that in basal conditions, NRF2 has very low activation levels without an electrophilic or pro-oxidant stimulation, therefore we attempted to tip the balance using only “NRF2 activated” datasets. The three cell lines included NRF2 activated (chemical or genetic) human airway cells (A549, BEAS-2B) and lymphoblastoid cells (LCL). Although all human based, there will be cell type or tissue specific differences for NRF2 targets in lung, blood cells, and liver, however we cast the widest net possible to include all detected ChIP-based targets in this analysis. The resultant comparison to liver-centric NRF2 targets may result in a higher rate of “false negatives”, and may have seen a higher validation of genes using a liver-based model. This point is now mentioned in the results, page 11, lines 405-408. Given our resource limited environment, we had to use previously published data. The additional analysis of detecting an ARE in these regions signifies that these are likely bone fide target genes. Indeed, some concerns of genome pull down due to complex protein interactions and 3D conformational chromatin interactions may result in NRF2 enrichment in genomic regions that NRF2 is not directly mediating transcription. To more clearly define likely cis-acting transcriptional mediation, we re-ran these datasets, limiting the NRF2-bound loci to just 10K bp with the location of bi-directional promotors using a new feature in “ChIPpeakAnno”. The combination of ARE identification + ChIP-identified loci within in promoter regions give the best possible evidence for a gene to be directly targeted (and presumably transactivated) by NRF2. 

How do the authors linked the experimental ChIP-seq regions to the biomarker gene set? May be a motif analysis of a "defined" promoter-region is more meaningful here. Perhaps this analysis was done already (see link: supplemental file 1 and line 390 in the manuscript) but may be the authors can add more details here. 

The original analysis simply looked for the closest gene region to the annotated peaks. Based on you and Reviewer 2’s suggestions, we limited the loci to those that are likely cis-acting (e.g. within 10Kbp of bi-directional promoter regions of genes and other transcribed regions). The combination of NRF2 bound peaks close to annotated gene promoters and the identification of bone fide ARE sequences within these peaks gave the best available evidence that these gene promoters are likely directly bound (and presumably regulated) by NRF2. This change is now noted on pg. 7, line 262.

In this paper the ChIP-seq analysis was done based on Ensembl database version 86 (Okt 2016). Why did the authors use this older version of the database in this actual paper?

For this re-run of the data, the current UCSC transcriptional database, last updated Oct 2019, was used to annotate the ChIP-seq peaks. This is noted on pg. 7, lines 263-264.

In figure 4 the term 'bioset' is used in the legend and in the paper. In the paper the term 'biomarker gene set' is used before. Do these two terms refer to the same gene set? A minor point but I was just confused about this two terms. May be it is a frequently used term, which I just do not know.

We are sorry this was not made clear. A bioset refers to a list of differentially expressed genes resulting from a specific comparison in the BSCE such as a chemical treatment vs control treatment. Text has been added to the first paragraph of the Methods to define this term.

Dose-, time-, and mutation-dependent NRF2 modulation:

The authors compare different chemicals screened in the ToxCast and/or Tox21 studies and relate these results to NRF2. First liver-cells are shown but also fibroblast and breast cancer cell line data are shown. The authors validate their findings using ARE-linked reporter system in HepG2 cells. The comparison with non-liver-related results is confusing. How comparable the findings between different cell types. Is there a possibility to estimate the completeness of this gene set?

The reviewer brings up some important points. The biomarker was constructed in a way to use in evaluating liver cell line data but also to use with transcript profiles derived from other cell lines that express Nrf2. Thus, we used a number of biosets from liver and non-liver cells to construct the biomarker to find common genes. Non-liver cells were used in exploratory data analysis when liver cell data was not available. Lung fibroblasts and MCF cells are known to express Nrf2 and respond to oxidative stress, and while the biomarker was not formally tested for accuracy in lung cells, it is not surprising to see activation in these comparisons. The inclusion of the breast cancer cell line was based in part on the fact that suppression of Nrf2 signaling by siRNA was carried out in this cell line. In answer to the question about the completeness of the gene set, we do not have an answer to that. It is very likely that there are other Nrf2 regulated genes that were not identified but that they may be more context specific. For example, there may be genes that are regulated by Nrf2 in chemical-specific and cell-specific manners as we note in the manuscript. However, the goal of the project was not to evaluate the Nrf2 regulon in multiple tissues but to find a set of genes consistently altered when Nrf2 is perturbed that can be used for prediction. We are confident based on our findings that the biomarker that we constructed is able to predict Nrf2 activation in many cell types. We hope this addresses the questions.

-----

Figure 1: Biomarker construction and screening strategy.

-> In all other Figure description the authors use Figure X. Description of the figure.

May be the ":" is a typo.

Thank you for noticing this. This has been fixed.

short comment to Figure 6:

May be the used red color should be defined in the legend of this figure.

A sentence has been added to the figure legend to describe the red shaded results.

Reviewer #2: The manuscript describes a very reasonable approach to using chemical perturbation transcriptional signatures to identify chemicals altering the activity of the transcription factor NRF2. I have several comments that I would like to see addressed by the authors.

Procedure used to construct the list of differentially expressed genes used as NRF2 biomarkers - Relevant sections: Methods (162-177), Results (334-338):

- I think some key details are missing from the methods. In general, the level of detail and tone of the methods section seems more adequate to the results section. Specifically:

- can you be more specific, and provide the logical rules that were used to construct these gene-sets, e.g. fold change ≥ 1.5 and adjusted p-value < ... in at least N/M NRF2 activation data-sets, and fold change < ... and adjusted p-value < ... in ... NRF2 activation data-set(s)?

Thank you for pointing out that this area needs further clarification. We have now added details in the Methods for describing how we filtered the genes that ultimately ended up in the biomarker. These include a fold change cutoff of >=1.5 averaged across all 7 biosets used to construct the biomarker in which Nrf2 is activated. The unadjusted p-value cutoff of p<0.05 that is used is inherent to the BaseSpace Correlation Engine data processing (see comments from reviewer #1). For genes to be included in the biomarker, expression changes had to be in the same direction in 4 of the 7 activation biosets, and none could include changes in the opposite direction. Lastly, the direction of regulation had to be in the opposite direction in the Nrf2 siRNA bioset. We have now added text in the Methods and the Results to better describe the criteria used to build the final biomarker used in the study.

- how was gene differential expression calculated? Please state what method was used (e.g. limma) and if a linear model was used, how were the experimental factors modelled? And what multiple test correction method. This applies to the analysis of other differential expression data-sets

A paragraph has been added to the methods section describing the gene expression workflow used in BaseSpace Correlation Engine. The methods used by BSCE to identify genes are deliberately less stringent than other approaches in that no multiple test correction is implemented. However, this works to our advantage when building a biomarker as genes that would be filtered out by the MTC are still available to be considered as candidates for incorporation into the biomarker. Our stringent cutoffs implemented in the biomarker construction procedures greatly reduce the number of possible Nrf2 regulated genes to find those that are cell line and chemical agnostic allowing the identification of a set of genes that are predictive of Nrf2 modulation.

- can you please specify if all genes were probed or only a subset? (some older microarray platforms only probed a subset of genes) 

This is an important point. BSCE only incorporates data from studies that use genome-level gene expression microarrays or RNA-Seq. With that said, there are examples of microarrays that do not cover the entire genome including the U133A Affy chip. Each GEO or ArrayExpress number for each bioset is provided in Supplemental File 1 to allow answering further questions about each of the biosets if the reader so desires.

- how different were these biomarker gene-sets from each other? E.g. can you show a reciprocal overlap matrix, and report their sizes.?

Performance assessment:

- I would appreciate a bit more detail on the performance for all ten biomarker gene-sets. You used ToxCast and Tox21 data to pick the best biomarker gene-set, therefore these data-sets were in a sense part of your training set. You don't have a completely separate test set, and so the performance may be a bit inflated. In general, it is a good practice to use cross-validation in this type of situation. However, if the performances of the ten biomarker gene-sets are all quite similar, I would not be too worried about this (also recognizing that you don't have a lot of ToxCast and Tox21 datasets, and so implementing cross-validation may be a bit tricky, the set is small, so things may get noisy). For the same reason, it would be useful to know how different were these biomarker gene-sets from each other (as commented above). You could use the ChIP-seq data to argue that the selection of the best biomarker set is robust (see comment below).

- Is this PPV really general? Usually the PPV can be different from sensitivity and specificity if the number of real positives is swamped by the number of real negatives (e.g. a test detecting a disorder that's present in 0.1% of the population). You have 14 real positives and 45 real negatives, does this reflect the general distribution in reality, can you comment on this?

The reviewer asks some very relevant questions about our analysis. We realize that this is confusing and detracts from the presentation of the value of the final biomarker used. In response to these concerns, we have deleted the text mentioning the other biomarkers as these do not add to the paper. We feel that the manuscript is long enough to describe the process of choosing the biomarker studied. We have also changed Figure 1 accordingly. These modifications now make the paper consistent with our past publications including one in PLOS ONE in which we focus on the final biomarker studied, the criteria for filtering genes and accuracy determination as we have done for the biomarkers for the estrogen receptor, the androgen receptor, and MTF1. 

In regards to the questions about prediction. First, we did not use the biosets used in making the biomarker in our prediction study. That is we did include the “training set”. Second, we had to use the Tox21/ToxCast data to provide a list of the true positives and negatives and we feel that this was the best dataset that could be used. These assays were trans-activation assays and did not involve gene expression profiling so including these identified chemicals examined in the same cell line but in different labs and likely under different exposure conditions provided a good test set for testing predictive accuracy. The reviewer makes a good point about the test was not balanced. Even in cases where the true positives and true negatives are not approximately equal, we decided to use the information we had to carry out the predictive accuracy.

Regarding the point about using the ChIP-Seq data to help guide the selection of the genes, it could be possible, but we have not tried it. We feel that if we use the ChIP-Seq data to guide selection of the genes, we may end up with genes that are not necessarily good for prediction of Nrf2 activation when you query microarray profiles. However, looking for overlap after selection of the genes helps to validate that we have selected a set of genes enriched in those that are directly regulated by Nrf2.

Regarding the comment about the distribution of Nrf2 positives and negatives in the chemical universe. Not sure anybody can answer that question with available data as it may likely be dependent on cellular context. However, screens for Nrf2 activation in the trans-activation studies could be used to answer that question. Our database is not set up to systematically answer that question due to the inherent heterogeneity of the dataset. Interesting question though.

ChIP-seq and PWM analysis:

- the percentage of signature gene with a ChIP-seq mark is impressive (68.5%). I would like a bit more details. For the annotation using ChIPpeakAnno, how far from a gene (transcription start or end) could a peak be? What fraction of genes had a peak before performing any intersection with the biomarker gene set? Can you show that this is much better as e.g. taking genes that were upregulated in at least one NRF2 activation data-set? Can you show that the top biomarker gene-set had also better ChIP-seq overlap (following up on the issue of potential inflation in performance) - similar considerations for the PWM analysis

Thank you for raising these concerns of potentially inflating the gene matches to the ChIP-seq datasets. As a result of this concern, we reevaluated these data to just annotate peaks within 10Kb of bi-directional promoter regions. This distance was dictated from our previous findings that many of NRF2 peaks with ARE regions fell within this range of gene transcriptional start sites (Chorley et al Nucleic Acids Research 2012). Using this cut-off, we enrich the possibility of finding NRF2-bound loci that are potentially cis-acting. The previous analysis identified features that were closest to the peak, however these distances could be 100kB + away. Although it is possible that conformational changes in the chromatin could create enhancer regions at these distances, we do not have the information to confirm this. Therefore, this re-evaluation enriched those loci near annotated human promoter regions. Disregarding the intersect with the biomarker gene set and using this new filter, this current analysis identified peaks 2819, 301 and 5212 transcriptional features near peaks. Combined, this is 7633 unique features. Note this is out of a possible 197,782 transcriptional features identified in the database. Of these features, 38 genes annotated with peaks were in common with the biomarker set. This lowered our match appreciably to 26.6%, however we more likely captured the genes that are being directly regulated by NRF2. Regarding enrichment of these peak-matched genes with other NRF2 data-sets, the purpose was not to validate that NRF2 was involved in the alteration of these genes (as would be captured by gene expression datasets, as suggested, and would be predictably a better overlap), but to show some supportive evidence that some of these biomarker genes were likely directly regulated by NRF2. While not of primary concern for the gene biomarker list generation (which could contain genes further downstream of target genes, i.e. non-direct), it does support that we are likely capturing some direct target genes of NRF2. Further refinement is achieved by demonstrating the presence of an ARE. With our new analysis we show that 19 of 38 genes with NRF2 peaks contain AREs (50%). This is a much better match than with the previous analysis that only found 21 ARE associated with 98 peak-associated genes (21.4%), supporting the idea that we are enriching regions with potential of direct NRF2 binding. These changes are now reflected in the updated Supplemental File 1 and on pg. 11, lines 403-412 and pg.15, lines 604-608.

Ingenuity pathway analysis

- can you please use multiple-test adjusted p-values? (or if you already did, make that explicit)

- can you provide more details on how this analysis is performed "top upstream regulating transcription factor that regulated the biomarker genes"

We have now performed a MTC using a Benjamini Hochberg analysis on the IPA results. While the more stringent cutoff resulted in some of the canonical pathways becoming insignificant, the upstream analysis results did not change. As such we have now reconstructed one of the figures and indicate in the Methods and legend that q-values were used.

Screening for chemicals that modulate NRF2

- can you explicitly comment on the anticipated false discovery rate? My back of the envelope calculation is: 9840 biosets * 1e-4 = 0.984 biosets, which is pretty stringent (this is very simplistic as it does not take into account correlations, which are probably abundant)

- for chemicals represents in multiple data-sets, how about using the median fold-change to collapse redundancy..?

The reviewer makes an interesting point here about the false discovery rate. In past studies with other biomarkers we tried to determine what p-value cutoff to use for the Running Fisher test. We started with a p-value of 0.001 and after a BH MTC the p-value cutoff was set at 0.0001 which we have used in all of our biomarker studies. Using this as our threshold for determining activation, our predictive accuracy has always been above 90% and ranges up to 98%. Importantly we are able to identify known activators which have correlation p-values close to 0.0001 for all of our biomarkers.

Regarding the point about collapsing the fold change values, we have not thought about that. We are not sure we understand what the reason would be for doing that. We feel that each bioset represents a unique exposure situation even if the same chemical is used. By not collapsing, we identify chemical exposure conditions that lead to activation. By collapsing gene sets from the same chemical, we could increase the false positive rate and false negative rates due to changes in the expression of different sets of genes within the biomarker. However, we can only speculate but we appreciate the idea that may be used in future studies if we can somehow develop code that could do this automated fashion.

Follow-up screening using the SRXN1-GFP assay

- in my mind, this experiment should be used as follows: from the results of 'Screening for chemicals that modulate NRF2', pick some that induce a positive effect, some that induce a negative effect, some that don't induce an effect; this selection should be unbiased, and it should exclude chemicals whose differential expression was used to construct and optimize the biomarker set (e.g. quercetin, sulindac, ...). Testing genes used to construct the biomarker gene-set would make sense only to validate this assay (e.g. it is known that quercetin activates NRF2). Doses should also be matched. Instead, it appears this experiment was designed to answer multiple questions (e.g. are negative results in the ToxCast and Tox21 used to select the best biomarker gene-set really negative, even when the biomarker gene-set suggests otherwise?), and it's harder to draw firmer conclusions; for instance, several of the chemicals use to construct and optimize the best biomarker gene-set were included.

- for these reasons, I am not supportive of this statement in the discussion << In an independent study

of 29 chemicals, the predictions based on the NRF2 biomarker were independently validated for most (81%) of the chemicals in HepG2 cells encoding a NRF2-responsive GFP reporter. >>

The reviewer brings up some important points here for this validation exercise. We see the point about including the two chemicals used to make the biomarker in the analysis. We have now recategorized them as positive controls and left them out of the analysis. Using the new list of chemicals, we have performed a predictive accuracy determination to find that the balanced accuracy is 78%. We have changed the text in the Results section and the Discussion regarding this analysis. 

Discussion

- it is quite long-winded and should be shortened/summarized a bit

We agree that there was some redundancy with the text in the Results. We have now shortened the discussion as suggested by cutting two paragraphs and removing additional text in other paragraphs. 

Other / General

- is it possible to have both effect size and p-value for biomarker signature correlations using Running Fisher..?

The reviewer brings up an intriguing point. If by effect size you mean in this case the number of overlapping genes that the correlation is calculated on or the fold-change of the genes, we have not determined that. The problem with fold-change is that the different microarray platforms have more compressed fold-change values compared to the profiles generated from RNA-Seq datasets. So the method used in CE, the fold-change rank based method compensates for these differences in fold-change between platforms. We have determined the effect of removal of different sets of biomarker genes in other studies (the estrogen receptor biomarker and TGx-DDI biomarker) and found that fewer genes generally results in lower sensitivity especially for genes that have higher fold-change (higher rank).

We have considered the -Log(p-value) as an effect in itself, a concept not universally appreciated. In a paper that was accepted in Toxicological Sciences this week from our lab, we have determined whether there is a -Log(p-value) that is associated with liver tumor induction in rats for a set of 6 gene expression biomarkers. In fact, we can identify these thresholds and use these for prediction. The basis for this is that as the factor is first activated, the number of genes altered that overlap with the biomarker is low and thus the correlation is low with the associated p-value close to 0.0001. As the factor is activated to a greater extent, there are more genes altered that overlap with the biomarker, a greater correlation and lower p-value, etc. So in our hands, with all of our biomarkers we see a relationship over and over again of greater activation leads to greater correlation and lower p-value. Thus, we determined that the p-value can be used to get an estimate of the level of activation of the factor and in the mentioned study, we found that when the p-value is low enough there is a level of activation of a factor (say AhR or CAR) that in a chronic situation would lead to liver tumors.

Table 1

- Is it possible to indicate the number of differentially expressed genes?

We have now added a column for the number of differentially expressed genes in each of the biosets.

Figure 2

- figure 2A has resolution issues, please remake it

The reviewer is correct in that the resolution is not optimal. We have remade the figure to improve resolution.

---

## [Decision Letter · Decision Letter 1]

11 Aug 2020

PONE-D-20-07833R1

Mining a human transcriptome database for chemical modulators of NRF2

PLOS ONE

Dear Dr. Corton,

Thank you for submitting your manuscript to PLOS ONE. After further consideration, we feel that minor revisions are required before the manuscript fully meet PLOS ONE’s publication criteria. Therefore, we invite you to submit a revised version of the manuscript that addresses the points raised  by one of the Reviewers during the second round of the process.

We look forward to receiving your revised manuscript.

Kind regards,

Roberto Mantovani

Academic Editor

PLOS ONE

Reviewers' comments:

Reviewer's Responses to Questions

**Comments to the Author**

1. If the authors have adequately addressed your comments raised in a previous round of review and you feel that this manuscript is now acceptable for publication, you may indicate that here to bypass the “Comments to the Author” section, enter your conflict of interest statement in the “Confidential to Editor” section, and submit your "Accept" recommendation.

Reviewer #2: (No Response)

2. Is the manuscript technically sound, and do the data support the conclusions?

Reviewer #2: Yes

3. Has the statistical analysis been performed appropriately and rigorously? 

Reviewer #2: Yes

4. Have the authors made all data underlying the findings in their manuscript fully available?

Reviewer #2: Yes

5. Is the manuscript presented in an intelligible fashion and written in standard English?

Reviewer #2: Yes

6. Review Comments to the Author

Reviewer #2: The manuscript is much clearer. Only minor revisions required.

OVERALL

The official gene symbol is 'NFE2L2'; but the synonym 'NRF2' has probably been used broadly. To avoid confusion, can you state this at the beginning of the manuscript, and use only one of these two names in the figures and tables? For instance, it is confusing to have in figure 3C a title 'Nrf2 upstream regulators' and then 'NFE2L2' listed as one (but it's actually the same as NRF2)

METHODS

<< The average fold-change across the 7 biosets in which NRF2 was activated had to be > |+/- 1.5-fold|. >> -- is this a log2 (FC)..? Please state explicitly

<< The p-value is the probability of the overlap between the NRF2 biomarker gene list and the IPA pathway gene list. The smaller the p-value the less likely that the association is random. >> -- can you adopt a more rigorous explanation?

RESULTS

Building the NRF2 biomarker and assessment of predictive accuracy

In the response to one of the reviewer comments the authors state << First, we did not use the biosets used in making the biomarker in our prediction study. That is we did include the “training set”. >> -- I think the authors meant "we did *not* include the “training set”"; can you make this clear in the results and methods?

Fig 2A: can the authors add labels with the name of the chemicals, or add a table with this information?

Characterization of the NRF2 biomarker genes.

<< Of the 143 NRF2 biomarker genes, 38 (26.6% ) of these were associated with the NRF2- bound ChIP-Seq loci near gene promoter regions (within 10 Kb) in at least one dataset derived from cell lines treated with NRF2-activating isothiocyanate, SFN, or have constitutively active NRF2 >> -- can you report what's the background rate, when considering all genes?

7. PLOS authors have the option to publish the peer review history of their article (what does this mean?). If published, this will include your full peer review and any attached files.

Reviewer #2: No

---

## [Author Response · Author response to Decision Letter 1]

19 Aug 2020

Reviewer's Responses to Questions

Comments to the Author

Reviewer #2: The manuscript is much clearer. Only minor revisions required.

We thank the reviewer for providing suggestions that helped to make this a better paper.

OVERALL

The official gene symbol is 'NFE2L2'; but the synonym 'NRF2' has probably been used broadly. To avoid confusion, can you state this at the beginning of the manuscript, and use only one of these two names in the figures and tables? For instance, it is confusing to have in figure 3C a title 'Nrf2 upstream regulators' and then 'NFE2L2' listed as one (but it's actually the same as NRF2)

We have now made a number of changes suggested by the reviewer. These include stating that Nrf2 is encoded by the NFE2L2 gene in both the abstract and introduction when Nrf2 is first introduced, changing NFE2L2 to Nrf2 in Fig 3C, and adding two changes in Table 1 to indicate that either the Keap1 or NFE2L2 gene are knocked down.

METHODS

<< The average fold-change across the 7 biosets in which NRF2 was activated had to be > |+/- 1.5-fold|. >> -- is this a log2 (FC)..? Please state explicitly

We have now changed the sentence to read “The average fold-change across the 7 biosets in which NRF2 was activated had to be ≥ |+/- 1.5-fold| (not Log2(fold-change).”. We hope this suffices.

<< The p-value is the probability of the overlap between the NRF2 biomarker gene list and the IPA pathway gene list. The smaller the p-value the less likely that the association is random. >> -- can you adopt a more rigorous explanation?

We removed the last sentence mentioned and added “Significant reported pathways have q-values < 1E-3.”.

RESULTS

Building the NRF2 biomarker and assessment of predictive accuracy

In the response to one of the reviewer comments the authors state << First, we did not use the biosets used in making the biomarker in our prediction study. That is we did include the “training set”. >> -- I think the authors meant "we did *not* include the “training set”"; can you make this clear in the results and methods?

Thanks for picking up this mistake. We have now changed the sentence on line 258 to read “The biosets used to create the biomarker (essentially the training set) were excluded from the test.” As suggested, we added the same sentence to the Results on line 409.

Fig 2A: can the authors add labels with the name of the chemicals, or add a table with this information?

We have now provided this information in Table S1. We now state on line 414 “Information about the biosets used in the test are provided in Table S1.”

Characterization of the NRF2 biomarker genes.

<< Of the 143 NRF2 biomarker genes, 38 (26.6% ) of these were associated with the NRF2- bound ChIP-Seq loci near gene promoter regions (within 10 Kb) in at least one dataset derived from cell lines treated with NRF2-activating isothiocyanate, SFN, or have constitutively active NRF2 >> -- can you report what's the background rate, when considering all genes?

We have now reported the background rate in the Results. The text now reads: “Of the 143 NRF2 biomarker genes, 38 (26.6% ) of these were associated with the NRF2-bound ChIP-Seq loci near gene promoter regions (within 10 Kb) in at least one dataset derived from cell lines treated with NRF2-activating isothiocyanate, SFN, or have constitutively active NRF2 [33, 38, 42]. This is in comparison to the background rate of 13.7%, where a total of 3553 genes contained NRF2-ChIP bound regions within 10 Kb of the TSS of the approximately 26,000 identified transcribed genes in the human genome (hg38). Therefore, the NRF2 biomarker genes are more significantly represented with NRF2-bound regions (Fisher’s Exact test, p<0.05).”

---

## [Editor Report · Decision Letter 2]

7 Sep 2020

Mining a human transcriptome database for chemical modulators of NRF2

PONE-D-20-07833R2

Dear Dr. Corton,

We’re pleased to inform you that your manuscript has been judged scientifically suitable for publication and will be formally accepted for publication once it meets all outstanding technical requirements.

Kind regards,

Roberto Mantovani

Academic Editor

PLOS ONE
---

## [Editor Report · Acceptance letter]

15 Sep 2020

PONE-D-20-07833R2 

Mining a human transcriptome database for chemical modulators of NRF2 

Dear Dr. Corton:

I'm pleased to inform you that your manuscript has been deemed suitable for publication in PLOS ONE. Congratulations! Your manuscript is now with our production department. 

Kind regards, 

on behalf of

Prof. Roberto Mantovani 

Academic Editor

PLOS ONE